# Human-Guided Fair Classification for Natural Language Processing

**Florian E. Dorner**[1,2]**, Momchil Peychev**[1]**, Nikola Konstantinov**[1]**, Naman Goel**[3]**,**
**Elliott Ash**[1]**, Martin Vechev**[1]
[1]ETH Zurich, [2]MPI for Intelligent Systems, Tübingen, [3]University of Oxford
Correspondence to: `florian.dorner@tuebingen.mpg.de`

## Abstract

Text classifiers have promising applications in high-stake tasks such as resume screening and content moderation. These classifiers must be fair and avoid discriminatory decisions by being invariant to perturbations of sensitive attributes such as gender or ethnicity. However, there is a gap between human intuition about these perturbations and the formal similarity specifications capturing them. While existing research has started to address this gap, current methods are based on hardcoded word replacements, resulting in specifications with limited expressivity or ones that fail to fully align with human intuition (e.g., in cases of asymmetric counterfactuals). This work proposes novel methods for bridging this gap by discovering expressive and intuitive individual fairness specifications. We show how to leverage unsupervised style transfer and GPT-3's zero-shot capabilities to automatically generate expressive candidate pairs of semantically similar sentences that differ along sensitive attributes. We then validate the generated pairs via an extensive crowdsourcing study, which confirms that a lot of these pairs align with human intuition about fairness in the context of toxicity classification. Finally, we show how limited amounts of human feedback can be leveraged to learn a similarity specification that can be used to train downstream fairness-aware models.

## 1 Introduction

With the rise of pretrained large language models (Sun et al., 2019), text classifiers can now be employed in tasks related to automated hiring (Bhatia et al., 2019), content moderation (Rieder & Skop, 2021) and social science research (Widmer et al., 2020). They are also part of machine learning pipelines for unsupervised style transfer (Reid & Zhong, 2021) or reducing the toxicity of language model outputs (Welbl et al., 2021). However, text classifiers have been shown to often exhibit bias based on sensitive attributes such as gender (De-Arteaga et al., 2019) or demographics (Garg et al., 2019), even for tasks in which these dimensions should be irrelevant. This can lead to unfair and discriminatory decisions, distort analyses based on these classifiers, or propagate undesirable demographic stereotypes to downstream applications. The intuition that certain demographic indicators should not influence decisions can be formalized in terms of the concept of *individual fairness* (Dwork et al., 2012), which posits that *similar inputs* should be *treated similarly* by machine learning systems. While in a classification setting similar treatment for two inputs can naturally be defined in terms of both inputs being labeled the same, the notion of input similarity should capture the intuition that certain input characteristics should not influence model decisions.

**Key challenge: generating valid, intuitive and diverse fairness constraints**   A key challenge when applying the individual fairness framework is defining the similarity notion $\phi$. Indeed, the definition is often contentious, as fairness is a subjective concept: what counts as a valid demographic indicator, as opposed to a problematic stereotype? Counterfactual definitions of similarity (Kusner et al., 2017) offer a principled solution, but they shift the burden towards the underlying causal model, whose definition can often be similarly contentious. While many other definitions have been proposed, it is widely recognized that the similarity of inputs can often be highly task dependent (Dwork et al., 2012; Barocas et al., 2019), e.g., two biographies that are identical except for indicators of gender may be considered similar in a professional context, but not in the context of online dating.

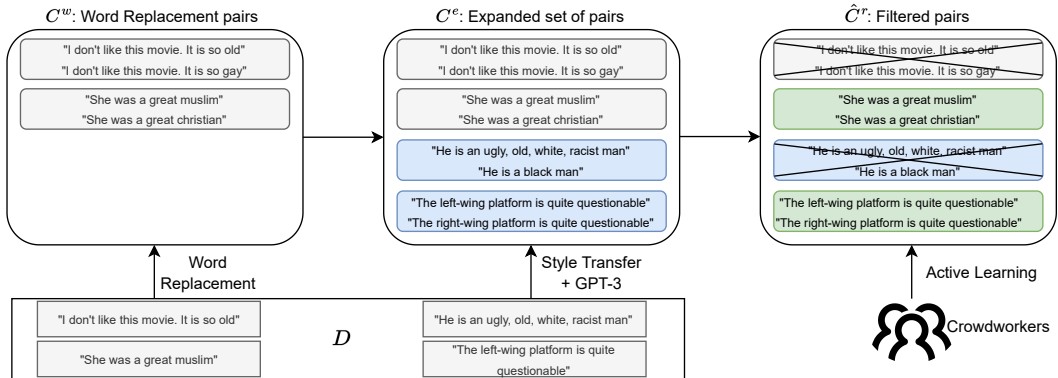

Figure 1: Workflow overview. We begin by generating sentence pairs using word replacement, and then add pairs of sentences leveraging style transfer and GPT-3. Then, we use active learning and crowdworker judgments to identify pairs that deserve similar treatment according to human intuition.

In the context of text classification, most existing works have cast similarity in terms of word replacement (Dixon et al., 2018; Garg et al., 2019; Yurochkin & Sun, 2021; Liang et al., 2020). Given a sentence $s$, a similar sentence $s'$ is generated by replacing each word in $s$, that belongs to a list of words $A_j$ indicative of a demographic group $j$, by a word from list $A_{j'}$, indicative of another demographic group $j' \neq j$. This approach has several limitations: (i) it relies on having exhaustively curated word lists $A_j$ of sensitive terms, (ii) perturbations that cannot be represented by replacing single sensitive terms are not covered, and (iii) many terms are only indicative of demographic groups in specific contexts, hence directly replacing them with other terms will not always result in a similar pair $(s, s')$ according to human intuition. Indeed, word replacement rules can often produce sentence pairs that only differ in an axis not relevant to fairness (e.g., by replacing "white house" with "black house"). In addition, they can generate so-called *asymmetric counterfactuals* (Garg et al., 2019): sentence pairs $(s, s')$ that look similar but clearly do not warrant similar treatment. For example, in the context of toxicity classification, the text "I don't like this movie. It is so old" may not be considered toxic while "I don't like this movie. It is so gay" clearly is.

**This work: generating fairness specifications for text classification** The central challenge we consider in this work is how to generate a diverse set of input pairs that aligns with human intuition about which inputs should be treated similarly in the context of a fixed text classification task. These pairs then induce fairness constraints that collectively define an implicit fairness specification on a downstream classifier, as individual fairness postulates that they should be classified in the same way.

We address this challenge via a three-stage pipeline, summarized in Fig. 1. First, we start from a training dataset $D$ for the text classification task under consideration and generate a set $C^w$ of candidate pairs $(s, s')$ by applying word replacement to sentences $s \in D$. Second, to improve diversity and expand on word replacement rules, we extend $C^w$ to a larger set of pairs $C^e$ by borrowing unsupervised style transfer ideas. We change markers of demographic groups, e.g., "women", "black people" or "Christians", in sentences $s \in D$ by replacing the style classifier used by modern unsupervised style transfer methods (Reid & Zhong, 2021; Lee, 2020) with a classifier trained to identify mentions of demographic groups. In addition, we add pairs from GPT-3 (Brown et al., 2020), prompted to change markers of demographic groups for sentences in $D$ in a zero-shot fashion. Third, to identify which of the generated pairs align with human intuition about fairness in the context of the considered classification task, we design a crowdsourcing experiment in which workers are presented with candidate pairs and indicate if the pairs should be treated similarly for the considered task or not. Since obtaining human feedback is expensive, we label a small subset of the generated pool and train a BERT-based (Devlin et al., 2019) classifier $\hat{\varphi}$ to recognize pairs that should be treated similarly, yielding a final set of filtered pairs $\hat{C}^r \subseteq C^e$. To further reduce labeling costs, we use active learning similar to (Grießhaber et al., 2020) to decide which pairs to label. We also demonstrate that the final set of constraints $\hat{C}^r$ can be used for training fairness-aware downstream classifiers, by adopting the Counterfactual Logit Pairing (CLP) regularizer of (Garg et al., 2019).

While our pipeline can in principle be used in the context of most text classification tasks, we instantiate it in the context of toxicity classification. Our experimental results, based on a large dataset for online content moderation, show that in this context our pipeline effectively generates a set of candidate pairs that covers more diverse perturbations than existing word replacement based approaches and successfully leverages human feedback to verify and filter these candidate pairs.

**Main contributions**    We make the following contributions:

- We introduce a method for generating datasets of diverse candidate pairs for individual fairness specifications. Towards that, we leverage GPT-3 and unsupervised style transfer to modify demographic attributes mentioned in sentences.
- We show that human feedback can be used for training a classifier that automatically identifies pairs that align with human fairness intuitions for a considered downstream task.
- We instantiate our framework in the context of toxicity classification. We experimentally show that the proposed pairs cover more diverse perturbations than word replacement, that crowdworkers agree with more than $75\%$ of proposed pairs and that our learned approximate specification can effectively be used to train fairness-aware downstream classifiers.

## 2    RELATED WORK

**Bias in NLP**    Early work on bias in Natural Language Processing has focused on unwanted correlations between the word embeddings of identifiers for protected demographic groups and unrelated categories such as occupations (Bolukbasi et al., 2016; Caliskan et al., 2017). More recently, generative language models have been found to harbor stereotypical biases (Liang et al., 2020; Nadeem et al., 2021; Vig et al., 2020; Smith et al., 2022). Specific to text classification, identity terms such as "gay" and explicit indicators of gender have been shown to significantly impact the outputs of classifiers trained to identify toxic comments (Dixon et al., 2018) or to predict a person's occupation from their biography (De-Arteaga et al., 2019). Olteanu et al. (2017) demonstrate that human perceptions of the quality of a toxicity classifier can depend on the precise nature of errors made by the classifier, as well as the annotators' previous experiences with hate speech. Similarly, Blodgett et al. (2020) recommend authors to explictly consider why, how and to whom the biases they identify are harmful.

**Language models for data augmentation**    Perez et al. (2022) use a language model to automatically generate test cases for another language model and Ross et al. (2022) automatically create contrast sets (Gardner et al., 2020) with a language model perturbing sentences based on control codes. Rios (2020) use style transfer to change the dialect of African-American Vernacular English tweets to Standard American English in order to evaluate the sensitivity to dialect of offensive language detectors, but do not extend style transfer to mentions of demographic groups. Hartvigsen et al. (2022) use language models to generate a balanced dataset of benign and toxic comments about minority groups and demonstrate that finetuning a toxicity classifier on this dataset can substantially limit its reliance on spurious correlations between identity terms and toxicity. However, their dataset is non-parallel, hence it cannot be used for evaluating individual fairness. Meanwhile, Qian et al. (2022) train a perturber model to imitate human rewrites $s'$ of comments $s$ that aim to modify mentions of demographic groups, and demonstrate that finetuning language models on the modified comments reduces demographic biases. Although this approach creates parallel data, it is limited by its reliance on large amounts of expensive human rewrites, which is likely why the authors only use it for perturbations along given demographic axes such as gender. In contrast, we allow for perturbations across axes and only require human annotations rather than rewrites.

**Learning fairness notions from data**    Ilvento (2020) provides an algorithm to approximate arbitrary individual fairness metrics for $N$ datapoints in $O(N \log N)$ queries, which can be practically infeasible. Meanwhile, Mukherjee et al. (2020) suggest training a classifier to predict binary fairness judgments on pairs $(s, s')$ in order to learn a fairness metric $\phi$, but restrict themselves to Mahalanobis distances on top of a feature representation $\xi(s)$, limiting their expressive power. In contrast to our work, these works do not validate their learned fairness notions with human feedback. To that end, Cheng et al. (2021) present an interface to holistically elicit stakeholders' fairness judgments, whereas Wang et al. (2019) aim to learn a bilinear fairness metric for tabular data based on clustering human annotations. Another strain of work aims to directly learn fair classifiers without an explicit

fairness metric: given access to similarity queries, Jung et al. (2021) propose an algorithm with generalization bounds for fairness and accuracy that requires polynomially many queries to cost-sensitive classification oracle, while other work (Gillen et al., 2018; Bechavod et al., 2020) focuses on online learning of individually fair models. Lastly, Lahoti et al. (2019) use examples of similar pairs $(s, s')$ to directly learn a representation that aims to ensure geometric similarity for similar pairs while preserving nearest neighbors in the input space. This approach is difficult to use for non-tabular data, in which nearest neighbor relations do not necessarily carry semantic meaning. In contrast to these works, we are not only interested in training fair classifiers, but also aim to learn the similarity function which approximates human intuitions about fairness for the task.

**Enforcing fairness constraints** Garg et al. (2019) suggest enforcing fairness constraints via censoring terms indicative of demographic groups, and by extending logit pairing (Kannan et al., 2018) to counterfactual logit pairing (CLP): during training, a classifier $f$ with logits $l$ is regularized by the term $\lambda||l(s) - l(s')||_2$ for similar datapoints $s$ and $s'$. Yurochkin et al. (2019) and Yurochkin & Sun (2021) use distributionally robust optimization and transport-based regularization respectively to train a toxicity classifier with distributional fairness guarantees for bilinear fairness metric similar to the ones from (Mukherjee et al., 2020). (Ruoss et al., 2020; Yeom & Fredrikson, 2021; Peychev et al., 2022) not only enforce, but also certify the adherence to individual fairness constraints expressed in logical formulas, weighted $L^p$ metrics or similarity sets defined in the latent space of a generative model. However, except for CLP and censoring, all of these methods require a known similarity metric with a specific functional form, which is not always available in practice.

# 3 METHOD

This section presents our end-to-end framework for generating candidate pairs for individual fairness specifications for a given text classification task, identifying candidates that indeed represent fairness constraints for that task and using them for training individually fair downstream classifiers. In Sec. 3.1 we expand on existing word replacement definitions of individual fairness in text classification by generating further candidate constraints. Next, in Sec. 3.2 we leverage human feedback to learn an approximate similarity function $\hat{\varphi}$ to identify a set of relevant constraints $\hat{C}^r \subseteq C^e$. Finally, in Sec. 3.3 we train a fairness-aware classifier $f$ using CLP on the filtered constraint set $\hat{C}^r$.

## 3.1 EXPANDING FAIRNESS CONSTRAINTS

We expand the word replacement based constraint set from (Garg et al., 2019) by implementing three different ways to modify markers of demographic groups mentioned in a sentence $s$: an extended word replacement list, unsupervised style transfer, and zero-shot modification using GPT-3.

**Word Replacement** First, we enrich the word replacement method by using the extensive lists of words associated with different protected demographic groups presented in (Smith et al., 2022). The pool of terms is substantially larger than the 50 identity terms from (Garg et al., 2019). We modify markers of group $j$ in a comment $s$ by replacing all words on the respective list of words associated with group $j$ with words from the list associated with the target group $j'$.

**Unsupervised Style Transfer** Second, we use an unsupervised style transfer approach based on prototype editing (see (Jin et al., 2022) for an extensive review on style transfer) to transform markers of a demographic group $j$ in a sentence $s$ to markers of another demographic group $j'$, creating a new sentence $s'$. Prototype editing identifies markers $a$ of a source style $A$ in a text $s$, and substitutes them by markers $a'$ of a target style $A'$. It can achieve unsupervised style transfer with minimal modifications to a source sentence $s$. Importantly, modern prototype editing algorithms rely solely on a style classifier to define their notion of style, so that they can transfer mentions of demographic groups when used with a classifier trained to identify such mentions.

Our approach consists of three phases. First, we train a multi-headed RoBERTa-based (Liu et al., 2019) classifier $c$ to predict the presence of mentions of demographic groups $j$ in a sentence $s$. Second, following (Reid & Zhong, 2021), we train a BART-based (Lewis et al., 2020) group-conditioned generator $g(s_t, j)$: given a sentence $s$ consisting of $n$ tokens that mentions group $j$, we remove mentions of demographic groups from $s$ by masking tokens at positions $k$ with above-average

attention weights $a_k \geq \bar{a}$, where $a_k$ represents the maximum attention weight at position $k$ in the penultimate layer of $c$ and the average is taken over all token positions for the sentence $s$. After merging consecutive masks, this yields a template $s_t$ on which $g(s_t, j)$ is trained to reconstruct $s$. Third, we modify sentences $s$ that mention group $j$ to instead mention group $j'$ by first creating a template $s_t'$ as in (Lee, 2020): we iteratively mask the tokens in $s$ for which masking most reduces the likelihood $p_c(j|s_t')$ of $j$ according to the group-presence classifier $c$, until it falls below a fixed threshold $T$. Then, we generate $s'$ as $g(s_t', j')$ using beam search with width 5 and selecting according to $p_c(j'|s') - p_c(j|s')$, the difference in likelihoods assigned to $j'$ and $j$ for $s'$ by $c$.

We use this approach rather than the attention-based masking from (Reid & Zhong, 2021) for the third step because the attention values $a_k$ are shared between the prediction heads of $c$ for all groups $j$. This means that attention-based masking might mask tokens related to a third group $j''$ instead of tokens related to $j$ for sentences $s$ in which multiple demographic groups are mentioned. While unlikely to be very detrimental during the training of the class-conditioned generator $g$, using attention for creating templates $s_t$ for $g$ can thus cause group transfer to target the wrong source group $j$.

The unsupervised style transfer approach promises multiple advantages. First, style transfer is likely to reproduce terms encountered during training, helping it to pick up on rare demographic terms that are particular to its training distribution which can be chosen to equal the training distribution for downstream tasks. In addition, unlike concurrent work by Qian et al. (2022), unsupervised style transfer only requires labels $y_j(s)$ indicating the mention of demographic group $j$ in a sentence $s$ rather than large amount of expensive human-produced examples of demographic group transfer. This allows us to modify mentions of demographic groups across axes like gender, religion and race, rather than restricting ourselves to changes within each of these axes.

**GPT-3**    Lastly, we make use of GPT-3 (Brown et al., 2020) to transform markers of protected demographic groups in a zero-shot fashion. We use three methods for generating pairs based on GPT-3. First, we prepend an example $s$ mentioning group $j$ with the prompt "Please rewrite the following sentence to be about $j'$ rather than $j$". Second, we use GPT-3's edit mode[1] with a similar prompt. Lastly, we generate candidate modifications $s'$ by word replacement, and postprocess them using GPT-3's edit mode with the prompt "Fix grammatical errors and logical inconsistencies".

While the GPT-3 approach does not automatically adapt to the relevant distribution of demographic terms, it does not require any additional data, or training of language models. To ensure that mentions of demographic group $j$ were indeed replaced by $j'$ going from $s$ to $s'$, we use the same group-presence classifier $c$ as for the unsupervised style transfer approach to heuristically identify successful group transfer and discard pairs $(s, s')$ for which group transfer failed, for all three of our approaches. Implementation details are described in App. B and App. E contains examples of generated sentences.

## 3.2    Learning the similarity function

In order to evaluate to what extent the proposed similarity criteria align with human intuition, we leverage human feedback, via a crowdsourcing study described in more detail in Sec. 4, to obtain labels $\varphi(s, s')$ which indicate whether the pair $(s, s')$ should be treated similarly for the sake of individual fairness ($\varphi(s, s') = 0$) or not ($\varphi(s, s') = 1$). In particular, identifying which pairs align with human labelers' intuition about fairness can help detect asymmetric counterfactuals, as well as failed attempts at style transfer for which $s'$ cannot be interpreted as a meaningful modification of $s$.

Since labeling all of $C^e$ can be prohibitively expensive, we train a probabilistic model $p_{\hat{\varphi}}(s, s')$ on a labeled subset of $C^e$ and use it to predict $\varphi(s, s')$ for the remaining pairs $(s, s')$. The similarity function $\varphi$ that specifies individual fairness is then approximated as $\hat{\varphi}(s, s') := 1 \Leftrightarrow p_{\hat{\varphi}}(s, s') > t$ for a given classification threshold $t$. Instead of using bilinear logits based on features for both $s$ and $s'$ (Mukherjee et al., 2020), we tokenize $s$ and $s'$ and train a BERT-based classifier on the concatenated tokens. This allows for a more holistic comparison between $s$ and $s'$ as attention heads can directly attend to differences between $s$ and $s'$ in earlier layers (see App. C for more details).

**Active Learning from human fairness judgments**    To make optimal use of costly human queries, we employ active learning when training the classifier $\hat{\varphi}$. We use the variation ratios $1 - \max_y p(y|x)$ to select the data points with the largest uncertainty about the correct label, an approach that is often

---

[1] https://openai.com/blog/gpt-3-edit-insert/

dubbed Least Confidence (LC) and estimate $p$ using a Dropout-based Monte-Carlo estimate (Gal & Ghahramani, 2016; Gal et al., 2017). As in (Grießhaber et al., 2020), we aim to save resources by precomputing features for the BERT-part of $\hat{\varphi}$ and performing Monte-Carlo dropout on the classification head of $\hat{\varphi}$ only. Concretely, after training $\hat{\varphi}$ on an initial randomly selected dataset $D_0 \subset C^e$ with labels $\varphi(s, s')$, we iteratively select new unlabeled training data $D_i \subset C^e \setminus \bigcup_{j<i} D_j$ with $|D_i| = 1000$, based on the variation ratios, query labels for $D_i$, and retrain $\hat{\varphi}$ on $D_i$. As different annotators can disagree about whether or not two sentences $s$ and $s'$ should be treated similarly, we use a majority vote for evaluation. Inspired by Chen et al. (2022)'s approach for dealing with noise in crowdsourcing, we use a single human query per pair $(s, s')$ during active learning, and relabel pairs that are especially likely to be mislabeled after active learning has concluded.

### 3.3 Training a fair(er) classifier

Finally, we train a fairness-aware classifier by accounting for the constraints defined by the learned similarity function. Specifically, we define the filtered constraint set $\hat{C}^r = \{(s, s') \in C^e : \hat{\varphi}(s, s') = 0\}$. We then train a RoBERTa-based (Liu et al., 2019) downstream classifier $f$, empirically enforcing the constraints implied by $\hat{C}^r$ by using the Counterfactual Logit Pairing (CLP) regularizer $\lambda \sum_{s,s':\phi(s,s')=0} ||l(s) - l(s')||_2$ of Garg et al. (2019). Here, $l$ represents the logits of the classifier $f$. If $\hat{\varphi}$ accurately approximates human fairness intuitions, this approach avoids enforcing constraints implied by asymmetric counterfactuals $(s, s')$ (pairs with $\varphi(s, s') = 1$) while properly enforcing actual constraints (pairs with $\varphi(s, s') = 0$). Further training details can be found in App. D.

## 4 Experiments

In this section, we experimentally evaluate our framework in the context of toxicity classification. Our key findings are: (i) the pairs generated by our method cover a wider range of perturbations compared to word replacement pairs only (Sec. 4.2), while mostly aligning with human intuition about individual fairness in toxicity classification (Sec. 4.3); (ii) the underlying similarity function $\varphi$ can be approximated by active learning from human judgements (Sec. 4.4), and (iii) the produced constraints can be used to enforce individual fairness on a downstream toxicity classifier (Sec. 4.5).

### 4.1 Dataset and setup

We focus on toxicity classification on the Jigsaw Civil Comments dataset[2]. The dataset contains around 2 million online comments $s$, as well as labels $toxic(s)$ indicating the fraction of human labelers that considered comment $s$ toxic. We define binary classification labels $y(s) := toxic(s) > 0.5$. A subset $D$ of the Civil Comments dataset also contains labels $A_j(s)$ that indicate the fraction of human labelers that think comment $s$ mentions the demographic group $j$. We again define binary classification labels as $y_j(s) := A_j(s) > 0.5$ for these comments, and use them to train our group-presence classifier $c$. We only consider the subset $D' \subset D$ for which no nan-values are contained in the dataset, and the RoBERTa-tokenized version of $s$ does not exceed a length of 64 tokens. We furthermore split $D'$ into a training set containing 75% of $D'$ and a test set containing the other 25%.

To build the pool $C^e$ of candidate pairs, for word replacement and style transfer, we attempt to produce modified comments $s'_{j'}$ mentioning group $j'$ for each $s \in D'$ for all demographic groups $j$ with $y_j(s) = 1$ and all possible target groups $j'$. For GPT-3, we use a subset of $D'$ due to limited resources. We then combine 42,500 randomly selected pairs $(s, s')$ with $s$ in the training part of $D'$ for word replacement and style transfer each and a total of 15,000 pairs $(s, s')$ for our three GPT-3 approaches, to form the set of candidate constraints $C^e$. We similarly construct a set of test constraints of a fourth of $C^e$'s size from the test portion of $D'$. More technical details can be found in App. B.

Throughout this paper, whenever we report individual fairness for a classifier, we refer to the proportion of pairs $(s, s')$ in a test pool of similar pairs for which $f(s) = f(s')$ rather than $f(s) \neq f(s')$. This metric captures that every instance of treating similar pairs differently is harmful. It is also a natural upper bound for certified individual fairness (Ruoss et al., 2020; Peychev et al., 2022)

---

[2]https://www.kaggle.com/competitions/jigsaw-unintended-bias-in-toxicity-classification/data

and prediction consistency (Yurochkin & Sun, 2021) which consider equal predictions across all comments $s'$ similar to a given comment $s$ for simpler formal specifications of similarity.

## 4.2 COVERAGE OF GENERATED FAIRNESS CONSTRAINTS

To validate that our approach covers a wider range of perturbations than word replacement, we train 4 different toxicity classifiers, using CLP with different constraint sets: our full constraint set $C^e$, as well as $C_1, C_2, C_3$ of the same size as $C^e$. The pairs in $C_1$, also referred to as $WR_{50}$, were generated by word replacement based on the 50 identity terms from (Garg et al., 2019) [3]; the pairs in $C_2$, also referred to as WR, were generated by word replacement by using the larger list of terms of (Smith et al., 2022); and the pairs in $C_3$, also referred to as ST, were created by the style transfer method. We cross-evaluate the performance of 4 classifiers trained with CLP with $\lambda = 5$, using each of the constraint sets $C_i$ in terms of test-time individual fairness measured as the proportion of similar pairs $(s, s')$ for which $f(s) = f(s')$ in $C^e$ and each of the 4 constraint sets $C_i$, and balanced accuracy. Table 1 reports these numbers and the performance of a "baseline" model, trained without CLP.

The results in Table 1 indicate that each classifier achieves high individual fairness when evaluated on test constraint pairs corresponding to the constraints used during its training (*in italics*) but performs worse when evaluated on other constraint pairs. This indicates that enforcing the similarity notions corresponding to different pair-generation methods can produce substantially different classifiers and that the adherence to individual fairness does not perfectly generalize across our generation methods. We note that training with CLP on $C^e$ or our style transfer pairs $C_3$ does not just yield significantly improved constraint adherence on $C_3$, but also generalizes well to $C_1$ and $C_2$ (see the numbers in bold), without losing much downstream accuracy. More details can be found in App. B and D.

| Training/Evaluation | BA | $WR_{50}$ ($C_1$) | WR ($C_2$) | ST ($C_3$) | Full $C^e$ |
|---|---|---|---|---|---|
| Baseline (no CLP) | $88.4 \pm 0.1$ | $78.4 \pm 1.4$ | $81.3 \pm 1.5$ | $76.7 \pm 1.8$ | $78.5 \pm 1.5$ |
| CLP $WR_{50}(C_1)$ | $87.0 \pm 0.3$ | *$98.3 \pm 0.1$* | $89.1 \pm 1.9$ | $86.3 \pm 1.9$ | $87.3 \pm 1.8$ |
| CLP WR ($C_2$) | $87.2 \pm 0.1$ | $93.1 \pm 1.2$ | *$98.2 \pm 0.4$* | $90.5 \pm 1.7$ | $92.9 \pm 1.2$ |
| CLP ST ($C_3$) | $85.9 \pm 0.1$ | **$95.3 \pm 0.4$** | **$97.1 \pm 0.3$** | *$95.4 \pm 0.4$* | $95.5 \pm 0.3$ |
| CLP Full $C^e$ | $85.0 \pm 3.4$ | **$95.5 \pm 0.9$** | **$97.8 \pm 0.6$** | $94.9 \pm 0.9$ | *$95.7 \pm 0.8$* |

Table 1: Balanced accuracy and individual fairness evaluated on different $C_i$ (proportion of pairs $(s, s') \in C_i$ for which $f(s) = f(s')$) for a RoBERTa-based toxicity classifier $f$ trained with CLP using different constraint sets $C_i$ in each row. Results are averaged over 5 runs and $\pm$ indicates the difference from the bounds of a naive $95\%$ confidence interval assuming normally distributed errors.

## 4.3 RELEVANCE OF GENERATED FAIRNESS CONSTRAINTS

To validate that the fairness constraints we generated are relevant and intuitive, we conducted a human evaluation using Amazon's MechanicalTurk. Workers were presented with a pair $(s, s')$ consisting of a comment $s$ from the Civil Comments dataset, as well as a modified version $s'$ and asked about whether they believe that the two comments should be treated similarly. Treatment was framed in terms of toxicity classification for content moderation, ensuring that we verify the relevance of the learned notions relevant to this specific task. Workers were also asked whether the demographic group was transferred correctly from a given $j$ to a given $j'$ and whether the content of $s$ has been preserved in $s'$ apart from the demographic group transfer. Further details can be found in App. A.

We collected human feedback for a set $S$ containing a total of 720 pairs $(s, s')$ with 240 each produced by our style transfer approach, GPT-3 in a zero-shot fashion, and word replacement using the list from (Garg et al., 2019)[4]. The 240 pairs per method were split into 80 pairs for each of the axes male↔female, christian↔muslim and black↔white, half of which with $s$ mentioning the first demographic group and half of them with $s$ mentioning the second group. Each pair $(s, s')$ was shown to nine workers and responses aggregated by a majority vote. Table 2 reports how often workers affirmatively answered the three questions from the previous paragraph for different methods.

---

[3]We did not discard any pairs based on a classifier for $C_1$.

[4]We again did not discard pairs based on a classifier for these pairs.

| Generation Method | Different Treatment Unfair | Group Transfer | Content Preservation |
|---|---|---|---|
| Word replacement | 85.9 (97.5) | 89.3 (95.0) | 88.1 (100) |
| Style Transfer | 85.2 (96.2) | 79.2 (85.4) | 79.2 (91.2) |
| GPT-3 | 83.2 (93.7) | 81.9 (89.5) | 78.4 (87.9) |

Table 2: Crowdsourced answers to questions about comment pairs $(s, s')$ for different methods for demographic group transfer: do the two comments deserve similar treatment, was the demographic group successfully transferred, and was content preserved apart from that? The first number represents the percentage of the answer across all queries, while the second number (in brackets) represents the percentage of comment pairs for which the answer was the majority vote across 9 queries.

The results in Table 2 demonstrate that all three methods produce relevant fairness constraints, according to a majority of annotators. At the same time, the workers' feedback indicates that the methods were mostly successful at modifying the mentioned demographic group, and at preserving content. While word replacement generally performs better in terms of group transfer and content preservation, this only translates to a small advantage in terms of producing pairs that represent actual fairness constraints $(\varphi(s, s') = 0)$. See Tables A.1 and A.2 for more detailed results.

## 4.4 LEARNING THE SIMILARITY FUNCTION

Since labeling pairs through human feedback is costly, obtaining labels for all candidate pairs in $C^e$ can be prohibitively expensive. Therefore, we employed our active learning approach to efficiently train our classifier $\hat{\varphi}$ from relatively few human judgments, with the goal of using it to identify pairs that represent actual fairness constraints on the remaining pool of candidates. We conducted 6 steps of active learning with 1000 queries each, selected by the LC criterion. Failed queries were discarded, so that we ended up with 5490 labeled pairs $((s, s'), \varphi(s, s'))$. Details can be found in App. C.

We evaluate our learned classifier on a test set $T$ consisting of 500 randomly selected pairs from $C^e$ for which five annotators were asked to predict the American's fairness judgment. We labeled pairs based on whether the majority thought they should be treated similarly $(\varphi(s, s') = 0)$, or not $(\varphi(s, s') = 1)$. Because 78.8% of the pairs $(s, s')$ in $T$ represented fairness constraints $(\varphi(s, s') = 0)$, we report Balanced Accuracy (BA), in addition to standard accuracy (ACC) and the true positive and negative rates (TPR and TNR). Table 3 displays these metrics for classifiers resulting from our active learning method for different classification thresholds $t$ and with and without subsequent relabeling.

| Method | ACC | TNR | TPR | BA |
|---|---|---|---|---|
| Constant Baseline | 78.8 | **100.0** | 0.0 | 50.0 |
| Active Learning t=0.5 | $79.8 \pm 0.3$ | $97.2 \pm 0.3$ | $15.1 \pm 1.2$ | 56.1 |
| Active Learning + Relabel t=0.5 | $\mathbf{81.1 \pm 0.3}$ | $95.5 \pm 0.7$ | $28.6 \pm 2.2$ | 62.0 |
| Active Learning t=0.1 | $80.0 \pm 0.5$ | $95.2 \pm 0.7$ | $23.7 \pm 3.5$ | 59.4 |
| Active Learning + Relabel t=0.1 | $80.7 \pm 0.6$ | $93.0 \pm 0.9$ | $35.0 \pm 1.3$ | 64.0 |
| Active Learning t=0.01 | $78.7 \pm 1.1$ | $87.5 \pm 2.1$ | $45.7 \pm 1.8$ | 66.6 |
| Active Learning + Relabel t=0.01 | $78.3 \pm 0.7$ | $86.8 \pm 1.5$ | $\mathbf{46.6 \pm 2.5}$ | **66.7** |

Table 3: Performance (in terms of accuracy, true negative/positive rate and balanced accuracy) for similarity classifiers $\hat{\varphi}$ trained on human fairness judgments with and without relabeling, evaluated on the test set $T$ with different decision thresholds $t$. Results are averaged over 10 repetitions of training on the data labeled in the last step/the relabeled data. $\pm$ indicates the difference from the upper/lower bound of a naive 95% confidence interval assuming normally distributed errors.

We observe that $\hat{\varphi}$ performs substantially better than random, achieving BA of 66.7% when used with an aggressive classifier threshold $t$. Table 3 also validates our relabeling approach: after observing that our classifier was biased towards predicting $\hat{\varphi}(s, s') = 0$ on a held-out validation set, we collected two additional labels for 500 pairs $(s, s')$ for which both the human and the predicted label were

equal to zero, selected based on the LC criterion. The majority vote over all three annotators was $\varphi(s, s') = 1$ for $47\%$ of these pairs, showing that our approach correctly identified pairs that were likely to be mislabeled. Retraining our classifier on the updated majority votes also substantially increased TPR at little costs to TNR, especially for balanced classification thresholds $t$ close to $0.5$. According to a qualitative evaluation, many sentence pairs $(s, s')$ predicted to not represent fairness constraints $(\hat{\varphi}(s, s') = 1)$ had the words "boy" or "man" replaced by terms denoting identity membership. Such sentence pairs, like "You boys don't go fishing when you go on those vacations, do you?" and "You Hindus don't go fishing when you go on those vacations, do you?" were often not seen as fairness constraints by human annotators, as the use of the identity term can be read as mocking. $\hat{\varphi}$ also identified sentence pairs $(s, s')$ for which $s'$ was unrelated to $s$, that were sometimes produced by GPT-3, as not representing fairness constraints. Further results can be found in App. C.

### 4.5 TRAINING A FAIRER DOWNSTREAM CLASSIFIER

Lastly, we evaluate whether the pairs $\hat{C}^r$ obtained by filtering $C^e$ with $\hat{\varphi}$ (trained with relabeling, threshold $t = 0.5$) can help with learning an individually fair downstream classifier, by training a RoBERTa-based toxicity classifier $f$ using CLP with $\lambda = 5$. More details can be found in App. D. We train toxicity classifiers with CLP using constraint sets defined by word replacement ($C_1$ and $C_2$ as in Sec. 4.2) and using all of $C^e$, or the filtered version $\hat{C}^r$. Additionally, we train on a challenging set of constraints, $C^e_{adverse}$ that consists of $C^e$ and 10,000 adversarially selected pairs $(s, s')$ created by randomly selecting comments $s$ with toxicity label $y(s) = 1$ and randomly selecting comments $s'$ with label $y(s) = 0$ from $D$, and a filtered version $\hat{C}^r_{adverse}$ of $C^e_{adverse}$ using threshold $t = 0.5$.

We then evaluate these classifiers in terms of Balanced Accuracy (BA) and individual fairness on the resulting classifiers on the test set $T$. Table 4 shows that compared to word replacement our expanded constraint set $C^e$ consistently yields better adherence to human-validated fairness constraints at the cost of a small drop in BA. However, we do not find a clear improvement from using the filtered constraint set $\hat{C}^r$ over the full set of constraints $C^e$. We hypothesize that this is due to our classifier $\hat{\varphi}$'s limited True Positive Rate combined with $\varphi(s, s')$ equalling zero for most pairs $(s, s') \in C^e$ according to human annotators, such that even filtering with a perfect classifier $\hat{\varphi}$ might be of limited utility as most constraints in $C^e$ are indeed relevant. This is supported by our results for $C^e_{adverse}$, where filtering substantially improves BA. Further experiments can be found in App. D.

| Method | BA | Fairness ($T$) |
|---|---|---|
| Baseline no CLP | $\mathbf{88.2 \pm 0.4}$ | $82.1 \pm 2.1$ |
| CLP WR$_{50}$ ($C_1$) | $87.1 \pm 2.0$ | $92.8 \pm 0.9$ |
| CLP WR ($C_2$) | $87.2 \pm 0.2$ | $95.8 \pm 0.9$ |
| CLP Full constraint set $C^e$ | $85.9 \pm 0.3$ | $96.5 \pm 1.4$ |
| CLP Filtered constraint set $\hat{C}^r$ | $85.9 \pm 0.5$ | $\mathbf{97.4 \pm 1.1}$ |
| CLP $C^e_{adverse}$ | $71.1 \pm 17.4$ | $97.8 \pm 2.2$ |
| CLP Filtered $\hat{C}^r_{adverse}$ | $79.3 \pm 2.2$ | $98.7 \pm 0.6$ |

Table 4: Balanced accuracy and individual fairness (proportion of human-validated similar pairs $(s, s') \in T$ with $\phi(s, s') = 0$, for which $f(s) = f(s')$) for toxicity classifiers $f$ trained with CLP and different sets of similar pairs. Results are averaged over 5 training runs. $\pm$ indicates the difference from the upper/lower bound of a naive $95\%$ confidence interval assuming normally distributed errors.

## 5 CONCLUSION

We proposed a framework for producing expressive and intuitive specifications for individual fairness in text classification. We experimentally demonstrated that our pairs are more expressive than word replacement pairs and that most of the generated pairs were relevant in the context of toxicity classification according to human annotators. We also trained a classifier that automatically identifies relevant pairs and showed that our approach can improve the fairness of a toxicity classifier.

Our dataset of fairness judgments is available at `https://github.com/eth-sri/fairness-feedback-nlp`

## 6    Ethics Statement

Our human evaluation experiments involving workers from Mechanical Turk were reviewed and approved by the ETH Zurich Ethics Commission as proposal EK 2022-N-117. Workers on Mechanical Turk were warned that they might be shown offensive comments as part of our study and were able to opt out of participating in our study at any time. We also made sure that the per-task compensation was sufficiently high to result in a hourly compensation exceeding the US federal minimum wage. More details on our human evaluation experiments can be found in App. A.

While we believe that our results show that learning more precise fairness notions by involving human feedback is a very promising area of research, we caution against directly using the labels from our human evaluation study $\phi$ for evaluating fairness in high-stakes real-world applications of toxicity classification. First, our results show that there is substantial disagreement between different survey participants about which pairs $(s, s')$ require equal treatment by a fair classifier. While resolving these disagreements via a majority vote is a natural choice, other approaches may be desired in some contexts, for example enforcing equal treatment whenever at least one participant believes it is required or explicitly accounting for distributional information via jury learning (Gordon et al., 2022) or multi-annotator architecture (Davani et al., 2022). Second, our survey participants are geographically biased to the US and are neither direct stakeholders, nor experts in discrimination law and hate speech.

Given that our learning approach shows promising signs of being able to improve upon existing approaches to quantifying individual fairness despite large amounts of disagreement, which is likely to be less common for actual stakeholders and experts, we recommend using it in conjunction with fairness judgments provided by application-specific experts and stakeholders. The collection of additional application-specific fairness labels $\phi$ is especially important when our framework is applied to downstream tasks other than toxicity classification, for which human agreement with the relevance of generated pairs to individual fairness could theoretically be substantially lower than indicated by our study. In addition, the validity of collected labels should be reassessed periodically in order to account for shifts in linguistic, cultural and social contexts (Aroyo et al., 2019).

In addition, our generated pairs only cover perturbations along a fixed list of demographic groups such that fulfilling individual fairness on these pairs does not guarantee invariance to other forms of perturbations. Correspondingly, evaluation on these pairs can help to provide better evidence about the extent of demographic biases but not necessarily other forms of bias present in a text classifier.

We believe that it is exceedingly difficult to capture human intuitions about individual fairness in complex domains like text using simple rules. Hence, our work implicitly defines similarity based on human-labeled data, which can be difficult to interpret. To instill further in the derived specifications, future work could aim to extract more interpretable specifications from the human feedback, for example using techniques from explainable machine learning.

## 7    Reproducibility Statement

We provide code to reproduce our generation pipeline and our experiments on synthetic data, as well as our dataset of human fairness judgments at `https://github.com/eth-sri/fairness-feedback-nlp`. All of our experiments involving transformer language models use the huggingface transformers library (Wolf et al., 2020). Additional details on our human evaluation are provided in App. A.

### Acknowledgements

We would like to thank Dan Hendrycks, Mislav Balunović, Afra Amini and Dominik Stammbach for helpful comments and discussions during early stages of this work. We also thank the anonymous reviewers for their insightful comments and constructive feedback that helped to improve this paper.

Florian Dorner is grateful for financial support from the Max Planck ETH Center for Learning Systems (CLS) received during part of this work. Nikola Konstantinov's contributions to this publication were made possible by an ETH AI Center postdoctoral fellowship.

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

## A    FURTHER DETAILS ON HUMAN EVALUATION

In order to participate, workers had to live in the US and be above 18 years old in addition to being experienced with MechanicalTurk (having completed more than 5000 HITs[5] and having a good reputation (97% acceptance rate across all of the worker's HITs). Workers were warned about the potentially offensive content of some of the comments show in the study by the following statement: "Please note that this study contains offensive content. If you do not wish to see such content, please withdraw from the study by leaving this website." and were also told that they could withdraw from the study at any later point: "You may withdraw your participation at any time without specifying reasons and without any disadvantages (however, you will not get paid for the current HIT in case you withdraw before completing it)".

After encountering a high prevalence of bots, malicious workers or workers that fundamentally misunderstood our task instructions during pilot experiments, we had workers pass a qualification test by providing correct answers for nine out of ten queries $\varphi(s, s')$ for pairs that were hand-designed to have a relatively obvious correct answer. We validated these hand-designed pairs in a separate experiment, querying workers about $\varphi(s, s')$ for 11 pairs $(s, s')$, and asking them to verbally explain each of their decisions, paying a total of $1.83. We only included hand-designed pairs in the qualification test if at least eight out of ten workers produced the intended answer during this experiment, and no worker brought forward convincing arguments against this answer being correct.

Workers were paid $0.83 for a battery of 11 queries $\varphi(s, s')$ for the Active learning and the test set $T$ and first $1.20, later $1.30 for 11 queries about all questions presented in Table 2. In both cases, workers had to explain one of their answers verbally, and one of the queries was a attention check pair $(s, s')$ with a relatively obvious correct answer constructed in the same way as for the qualification tests.

Blocks of queries with wrong answers to the attention check question or (partially) incoherent verbal explanations were manually reviewed, and thrown out in case we were not able to find evidence that the worker had correctly understood the task in which case the worker was not paid. Workers were made aware of this by the following statement in the task description: "To ensure workers pay attention, we include pairs of comments for which there is a clear correct answer at random positions. Please make sure to carefully read every comment and answer diligently, as your HIT will be rejected if your answer for one of these pairs of comments is wrong, in which case you will not receive a payment for the HIT. We will also ask you to produce an explanation for one of your answers. Please write at least one sentence and make sure your explanation is coherent with the answer you gave, as your HIT might be rejected otherwise in which case you will not receive a payment for the HIT.". According to https://turkerview.com, a tool used by many professional workers on MechanicalTurk, we paid workers an average hourly rate of $16.17, clearly exceeding the US minimum wage. While this is likely an overestimate, as not all workers use turkerview, the hourly rate is so high, that we still exceed the minimum wage for workers taking twice as long as the average worker using turkerview.

Workers were introduced to the task as follows: "We are investigating fairness in toxic comment classification and are looking for pairs of comments with similar content that should be treated the same by a content moderator (if one of them was removed, the other one should be removed too). You will be shown pairs of comments and be asked whether the comments are similar in various ways and whether you would consider it unfair for a content moderator to remove one of the comments in each pair but not the other.". The queries read as follows for the first task:

- Without further context, would the average american consider it unfair or acceptable for a content moderator to delete one of the shown comments but not the other?
  - It would be unfair for one of the two comments but not the other to be deleted (Both comments ought to be treated the same way, no matter what)
  - If comment a) was deleted, comment b) should be deleted as well, but not the other way around. (Deleting comment b) would be more justifiable)
  - If comment b) was deleted, comment a) should be deleted as well, but not the other way around. (Deleting comment a) would be more justifiable)
  - It would be acceptable to delete either of the comments without deleting the other. (Treating the comments differently either way would be fine)

---

[5] Bundled tasks on MechanicalTurk for which a remuneration is received on completion

and as follows for the second one (Placeholders for group terms are presented in *Italics*. The answer summarized in Table 2 is highlighted in **bold**):

- Without further context, would you consider it unfair or acceptable for a content moderator to delete one of the shown comments but not the other?
  - **It would be unfair for one of the two comments but not the other to be deleted (Both comments ought to be treated the same way, no matter what)**
  - If comment a) was deleted, comment b) should be deleted as well, but not the other way around. (Deleting comment b) would be more justifiable)
  - If comment b) was deleted, comment a) should be deleted as well, but not the other way around. (Deleting comment a) would be more justifiable)
  - It would be acceptable to delete either of the comments without deleting the other. (Treating the comments differently either way would be fine)
- What do you expect the average American to answer to the previous question?
  - **It would be unfair for one of the two comments but not the other to be deleted (Both comments ought to be treated the same way, no matter what)**
  - If comment a) was deleted, comment b) should be deleted as well, but not the other way around. (Deleting comment b) would be more justifiable)
  - If comment b) was deleted, comment a) should be deleted as well, but not the other way around. (Deleting comment a) would be more justifiable)
  - It would be acceptable to delete either of the comments without deleting the other. (Treating the comments differently either way would be fine)
- Do you consider one of the two comments to be more factually correct than the other?
  - Comment a
  - Comment b
  - **Neither comment / too little context to tell**
- Do you consider one of the two comments to be more grammatically correct than the other?
  - Comment a
  - Comment b
  - **Neither comment**
- Is comment a) about *group a* and comment b) about *group b*?
  - **Yes**
  - No, comment a) is not about *group a*
  - No, comment b) is not about *group b*
  - No, neither
- Apart from differences related to *group a* and *group b*, are both comments similar in terms of content?
  - **Yes, they are almost the same.**
  - They are somewhat similar, but differ in some additional details.
  - There is an important additional difference between the comments' content

Table A.1 shows an extended version of Table 2 and includes human annotator's answers to additional questions. It shows that the reason why the advantages of word replacement in terms of group transfer and content preservation do not fully translate to producing pairs that represent actual fairness constraints could be due to its worse performance in terms of preserving factuality. Indeed, we found examples in which word replacement transformed "white house" to "black house"; or Obama is referred to as "white" rather than "black" in a modified comment. These pairs were not seen as fairness constraints by most annotators, while also being judged badly in terms of preserving factuality.

Table A.2 shows the results of the human evaluation on our test set $S$ split along the axis of attribute transfer, rather than generation method as in 2. Along with the results in Table 2 they show that despite the general agreement about the relevance of the generated fairness constraints, there is

| Metric/Method | Word replacement | Style Transfer | GPT-3 |
|---|---|---|---|
| Unfair: Average American | 84.9 (97.5) | 84.6 (95.8) | 83.4 (95.0) |
| Unfair: Own Opinion | 85.9 (97.5) | 85.2 (96.2) | 83.2 (93.7) |
| Group Transfer | 89.3 (95.0) | 79.2 (85.4) | 81.9 (89.5) |
| Content preservation | 88.1 (100) | 79.2 (91.2) | 78.4 (87.9) |
| Same Factuality | 73.0 (84.1) | 76.2 (87.5) | 78.5 (89.1) |
| Same Grammaticality | 91.2 (99.1) | 92.9 (97.9) | 92.9 (98.3) |

Table A.1: Human evaluation: Answers to questions about comment pairs $(s, s')$ grouped by different methods for demographic group transfer. The first number represents the fraction of the answer across all queries, while the second number (in brackets) represents the fraction of comment pairs for which the answer was the majority vote across 9 queries.

substantial disagreement between annotators when it comes to deviations from the most common answer across all comments. In all cases, the fraction of comments with majority vote equal to that answer is substantially higher than the overall fraction of these votes across all comments and annotators. The same is true for our set $T$ of 500 randomly selected pairs from $C^e$ for which we only asked about the average American's fairness judgment: 70.9% of the annotations were $\varphi(s, s') = 0$, while the same was true for 78.8% of the per-comment pair majority votes.

| Metric/Method | male↔female | black↔white | christian↔muslim |
|---|---|---|---|
| Unfair: Average American | 83.5 (96.6) | 82.2 (94.5) | 87.2 (97.0) |
| Unfair: Own Opinion | 83.5 (96.6) | 82.4 (92.9) | 88.4 (97.9) |
| Group Transfer | 82.6 (91.6) | 81.6 (86.6) | 86.2 (91.6) |
| Content preservation | 84.9 (95.4) | 79.5 (92.0) | 81.3 (91.6) |
| Same Factuality | 75.3 (82.9) | 73.6 (85.0) | 78.8 (92.9) |
| Same Grammaticality | 90.5 (97.5) | 92.2 (98.3) | 94.3 (99.5) |

Table A.2: Human evaluation: Answers to questions about comment pairs $(s, s')$ grouped along demographic group transfers along different axes. The first number represents the fraction of the answer across all queries, while the second number (in the brackets) represents the fraction of comment pairs for which the answer was the majority vote across 9 queries.

Our dataset including the pairs generated by our approach and aggregate human fairness judgments can be accessed at `https://github.com/eth-sri/fairness-feedback-nlp`.

## B  FURTHER DETAILS ON STYLE TRANSFER

All of our experiments involving transformer language models use the huggingface transformers library Wolf et al. (2020).

**Unsupervised style transfer**   To transform markers of demographic groups in sentences, we first finetune a Multi-headed RoBERTa-based (Liu et al., 2019) classifier $c$ to predict labels $y_j$ indicating the presence of markers of a demographic group $j$ from a list of protected demographic groups $J$ in a sentence $s$. We use the population labels ("Black", "Male", "Heterosexual", "Muslim", etc.) that are provided for a subset of the Civil comments dataset. The group-presence classifier $c$ is based on the roberta-base model, followed by a linear layer with 768 neurons applied to the output embedding of the first token only, a Tanh layer, another linear layer mapping to a single dimension, and a Sigmoid layer. We train $c$ for 3 epochs with a batch size of 16 and use the Adam optimizer Kingma & Ba (2015) with learning rate 0.00001 to optimize the binary Cross Entropy loss, reweighed by relative label frequency in the dataset. Table B.1 shows the balanced accuracy on the test set for all demographic groups in the dataset. For our downstream applications of $c$, we restrict ourselves to the demographic groups for which the classifier $c$'s balanced accuracy is above $90\%$. Furthermore, we also exclude the group labeled "mental illness" because the word replacement lists we used lack a clear analogon.

Then, we finetune a BART-based (Lewis et al., 2020) generator $g$ on a mask-filling task on the same data: For every data point $s$, we sample a group from the set of demographic groups $j$ mentioned in $s$, i.e. $\{j : y_j(s) = 1\}$, skipping sentences $s$ for which no group $j$ meets this criterion. Inspired by (Reid & Zhong, 2021) we mask all of $s$'s tokens that have an above-average attention value for the 11th layer of the classifier $c$, merge consecutive mask tokens into one, and prepend the name of the sampled group $j$ to the masked sentence before fedding it to the generator $g$. The generator $g$ is then finetuned to reconstruct $s$ using token-wise Cross Entropy.

The BART-based generator $g$ is trained starting from the pretrained facebook/bart-large model for a single epoch with batch size 4, again using Adam and a learning rate of 0.00001. For filling in masked sentences, we pick the completion with the largest difference in the classifier $c$'s pre-sigmoid activation for the target and source demographic groups $j'$ and $j$ among candidate sentences produced by a beam search generation using the generator $g$ with width 5.

To transfer an example $s$ from mentioning group $j$ to mentioning group $j'$, we follow (Lee, 2020) and iteratively mask the token for which masking reduces $p_c(y_j|x)$ the most, until we reach a threshold of $p_c(y_j|x) < 0.25$. We use this approach rather than the attention-based masking from (Reid & Zhong, 2021) because of the lack of theoretical motivation for using attention to identify important features (Bastings & Filippova, 2020), and because attention scores are the same for all of our model's group-presence prediction heads, rather than specific to a particular group $j$.[6] Then, we prepend a verbal representation of label $j'$ to $s$ to form a prompt $p$, and generate a sentence $s'$ as $g(p)$.

| Category | BA | Category | BA | Category | BA |
|---|---|---|---|---|---|
| Male | 96.5 | Christian | 96.6 | Physical disability | 54.9 |
| Female | 97.8 | Jewish | 98.9 | Intellectual disability | 54.3 |
| Transgender | 99.3 | Muslim | 98.9 | Mental illness | 98.3 |
| Other gender | 50.0 | Hindu | 98.2 | Black | 99.2 |
| Heterosexual | 98.1 | Buddhist | 99.2 | White | 99.5 |
| Homosexual | 99.3 | Atheist | 99.6 | Asian | 98.3 |
| Bisexual | 65.4 | Other religion | 50.0 | Latino | 96.6 |
| Other sexuality | 50.0 | Other disability | 50.0 | Other race | 55.5 |

Table B.1: Balanced accuracies of the group-presence classifier $c$ for different labels

**Word replacement**   Our word replacement approach is based on the list of words provided in Smith et al. (2022): Given a sentence $s$ mentioning demographic group $j$ and a target attribute $j'$, we replace

---

[6]We used attention during the training of $g$, for which dropping out some tokens unrelated to $j$ is less problematic, in order to save resources.

all words in $s$ that are on the list associated with $j$ with random words from the list associated with $j'$, replacing nouns with nouns and descriptors with descriptors whenever possible, and nouns with descriptors otherwise. The full list of words we used for word replacement is displayed in Table E.1.

**GPT-3**    We accessed GPT-3 using OpenAI's API[7]. For our first approach, we used the "text-davinci-001" version of GPT3 in a zero-shot manner with the prompt: "Please rewrite the following sentence to be about $j'$ rather than $j$:" followed by a new line and the targeted sentence $s$. The second approach was based on the beta-version of GPT-3's editing mode [8]. Here, $s'$ is produced using the model "text-davinci-edit-001" with the instruction "Rewrite the text to be about $j'$ rather than $j$". Lastly, we used to same model in conjunction with word replacement: First, we generated a candidate sentence $s''$ using the procedure described in the word replacement section. Then, in order to fix issues caused by the context-blindness of the word replacement approach, we postprocessed $s''$ using "text-davinci-edit-001" with the instruction "Fix grammatical errors and logical inconsistencies" to produce $s'$. We used temperature $= 0.7$ and top_p$= 1$ in all our approaches and used max_tokens$= 64$ for "text-davinci-001" to control the length of the modified sentence $s'$.

Please refer to the most up-to-date version of OpenAI's usage policy[9] regarding content generation using GPT-3.

**Post-filtering**    For all three approaches, we performed a post-filtering step to reduce the prevalence of unsuccesful attempts at demographic group transfer in our set of constraints $C^e$. Given a pair $(s, s')$ of an original sentence and a modified version, we only include it in our set of constraints $C^e$, if the classifier probability $p_c(y_{j'}|s')$ for label $j'$ is below $0.5$ and the classifier probability $p_c(y_j|s')$ for label $j$ is above $0.5$.

As mentioned in Sec. 4.1, we attempt to produce modified comments $s'_{j'}$ mentioning group $j'$ for each $s$ in $D'$ for all demographic groups $j$ with $y_j(s) = 1$ and all possible target groups $j'$ for word replacement and style transfer. For GPT-3, we attempted a total of 75 generations for each of our three generation modes per axis pair of demographic groups $(j, j')$ and direction of group transfer, with the source sentences $s$ randomly selected among the sentences with label $j$ in $D'$. For constructing the secondary test set $S$, we attempted more generations for the axes male↔female, christian↔muslim and black↔white, homosexual↔heterosexual. The latter axis was left out of $S$ because we found that the rate of successful generations was too limited. We generated a maximum of 2250 attempts up until a total of 250 successful generations (post-filtering step passed) for GPT-3's zero-shot mode, a maximum of 750 until to a total of 100 successful generations for GPT-3's edit mode, and up until a total of 100 successful generations for GPT-3 based postprocessing of word replacement. Table B.2 shows the overall amount of generated pairs per method.

| Generation Method | Total (Train) | Total (Test) | In $C^e$ (Train) | In $C^e$ (Test) |
|---|---|---|---|---|
| Word Replacement | 980667 | 331490 | 42500 | 10625 |
| Style Transfer | 681111 | 229883 | 42500 | 10625 |
| GPT-3 Zero-Shot | 6322 | 2139 | 6200 | 1550 |
| GPT-3 Edit Mode | 3704 | 1199 | 3500 | 875 |
| GPT-3 Postprocessing | 5330 | 1831 | 5300 | 1325 |

Table B.2: Amount of generated pairs $(s, s')$ per generation method.

As an additional experiment to validate the increased diversity of our constraint set $C^e$ we train a similarity classifier[10] $\hat{\varphi}$, on $C^e$ to distinguish pairs $(s, s')$ generated by word replacement from pairs generated by style transfer or GPT-3. Training on 100000 examples without label noise, we are able to achieve over $91.6\%$ test accuracy on a balanced test set, suggesting that there is a meaningful difference between pairs generated by word replacement and the rest of the constraint candidates $C^e$.

---

[7]https://openai.com/api/

[8]https://openai.com/blog/gpt-3-edit-insert/

[9](https://platform.openai.com/docs/usage-policies

[10]Using the same architecture as for our active learning experiments described in App. C

## C FURTHER DETAILS ON LEARNING SIMILARITY FUNCTIONS

First, Proposition C.1 below establishes that robustness with respect to a binary similarity function $\varphi$, i.e. $\varphi(s, s') = 0 \Rightarrow f(s) = f(s')$, can fully capture the definition of individual fairness as Lipschitz-Continuity proposed by Dwork et al. (2012) for deterministic classifiers $f$.

**Proposition C.1.** *Given a metric $d : X \times X \to \mathbb{R}$, a binary metric $d_b : Y \times Y \to \{0, 1\}$ and a constant $L > 0$, there exists a similarity function $\varphi : X \times X \to \{0, 1\}$ such that a function $f : (X, d) \to (Y, d_b)$ is Lipschitz-Continuous with constant $L$ if and only if $\varphi(x, x') \geq d_b(f(x), f(x'))$ for all $x, x' \in X$.*

*Proof.* Define $\varphi(x, x') := \mathbb{1}\{Ld(x, x') \geq 1\}$. Then whenever $d_b(f(x), f(x')) = 1$, we have $d_b(f(x), f(x')) = 1 \leq \varphi(x, x')$ if and only if $d_b(f(x), f(x')) \leq Ld(x, x')$. But if $d_b(f(x), f(x')) = 0$, the Lipschitz inequality is allways true. Now, assume that $f$ is not Lipschitz: Then, there exist $x, x' \in X$ such that $1 = d_b(f(x), f(x')) > Ld(x, x')$, implying $0 = \varphi(x, x') < d_b(f(x), f(x')) = 1$ □

We use a BERT-based classifier that acts on a pair $(s, s')$ by first tokenizing both $s$ and $s'$ and padding the token representation to a length of $64$, concatenating these tokens and feeding the concatenated token representation into a pretrained bert-uncased-base model. We then apply a linear layer with dropout ($p = 0.1$) followed by a Tanh layer and a second linear layer with dropout ($p = 0.1$) to obtain single dimensional logits, to which a sigmoid layer is applied before computing the binary Cross Entropy loss. We use BERT rather than more modern models such as RoBERTa (Liu et al., 2019) and Deberta (He et al., 2021), as we have found it to clearly outperform them for our task, plausibly because BERT uses a next-sentence-prediction task during pretraining, which is structurally similar to our task of comparing two sentences. Table C.1 demonstrates the advantage of using BERT, as well as concatenating token representations rather than learning based on the difference between separately produced BERT features for both $s$ and $s'$. Unless stated otherwise, our Active Learning approach trains for five epochs on each queried block $D_i$ before selecting new data $D_{i+1}$ to label. Example generations for our different methods can be found in App. E.

| Model | BA |
|---|---|
| BERT-Concat | **86.7** |
| BERT-Merge | 79.9 |
| BERT-Featurediff | 67.8 |
| DeBERTa-Concat | 54.7 |
| DeBERTa-Merge | 53.2 |
| DeBERTa-Featurediff | 50.8 |
| RoBERTa-Concat | 52.1 |
| RoBERTa-Merge | 50.3 |
| RoBERTa-Featurediff | 51.1 |
| BERT-Large-Concat | 84.4 |
| BERT-Large-Merge | 84.1 |
| BERT-Large-Featurediff | 59.2 |
| BERT-Bilinear | 50.7 |

Table C.1: Different architectures trained for one epoch on 5000 samples from a set of pairs $(s, s')$ generated using word replacement to distinguish demograpghic group transfer within the same category of gender and sexuality, race and religion vs across categories ($\varphi_2$). "Featurediff" uses a linear model applied to the difference of model features produced for the respective first tokens in $s$ and $s'$. "Bilinear" uses a bilinear model on top of these feature differences instead. "Merge" appends $s'$ to $s$ before tokenization and learns a linear model on top of the model features for this combined input. "Concat" operates similarly, but first tokenizes $s$ and $s'$ and pads both to $64$ tokens before feeding the concatenated tokens into the model. No dropout was used in the post-BERT layers for these experiments. All results averaged over 10 runs and $\pm$ indicates the difference from the upper/lower bound of a naive $95\%$ confidence interval assuming normally distributed errors.

## C.1 SYNTHETIC DATA

For active learning, we freeze the underlying BERT model during the active learning selection and only apply MC-Dropout on the level of the classifier head, similar to (Grießhaber et al., 2020), but unlike them we do not use BALD (Houlsby et al., 2011) and instead approximate $p(y|s, s')$ averaging the models' predicted probabilities $p_{\hat{\varphi}}(y|s, s', w)$ for 50 sampled dropout masks $w$. We call this approach LC-UNC and experimented with various alternative selection criteria. Unlike LC-UNC, LC directly approximates $1 - \max_y p(y|s, s')$ using a single forward pass through the $\hat{\varphi}$ with deactivated dropout. BALD is the approach from Grießhaber et al. (2020), while VARRA and Majority approximate $1 - \max_y p(y|s, s')$ using MC-Dropout differently than LC-UNC: In Majority, $p(y|s, s')$ is approximated as the fraction of dropout samples $w$ for which $\hat{\varphi} = 1$, while VARRA averages $1 - \max_y p_{\hat{\varphi}}(y|s, s', w)$ over dropout samples $w$ instead of averaging $p_{\hat{\varphi}}(y|s, s', w)$ before applying the maximum operator. In addition, the table contains the "automatic relabeling" condition in which $D_i$ is selected from the whole of $C^e$ rather than just the previously unlabeled examples $D_i \subset C^e \setminus \bigcup_{j<i} D_j$. During training, pairs $(s, s')$ that have been queried multiple times are labelled according to the majority vote of all queries, and as $0.5$ in case of a tie.

We validate the efficacy of our active learning approach for learning the similarity function $\varphi(s, s')$ with a limited amount of noisy queries. For this, we define two synthetic similarity functions $\varphi_i : i \in \{1, 2\}$. The first, $\varphi_1$ is equal to zero, whenever a pair $(s, s')$ was generated via word replacement and equal to one otherwise, as in the first experiment from the previous section. The second, $\varphi_2$ is equal to zero, whenever the group $j$ of $s$ that was removed and the added group $j'$ in $s'$ are within the same category of gender and sexuality, race, or religion, and equal to one otherwise. For example, a pair $(s, s')$ for which markers of "White people" in $s$ were modified to markers of "Black people" in $s'$ would have $\varphi_2(s, s') = 0$, while $\varphi_2(s, s')$ would be one if the group was modified to "muslim" in $s'$ instead. We simulate the label noise introduced by annotators' disagreement by independently flipping each label with probability $p = 0.3$ during training the similarity classifier $\hat{\varphi}$. For training with 3 instead of one query per data point, we reduce the overall amount of training data from 10000 samples in $C^e$ to 3333 samples and reduce the probability of flipping labels to $p = 0.216$, simulating a majority vote. In turn, the active learning approach selects 333 instead of 1000 data points for labeling in each of its ten steps in that scenario. Table C.2 shows that active learning noticeably outperforms randomly sampling data points for our task, that there is no clear direct benefit from employing multiple queries per pair $(s, s') \in C^e$ over obtaining labels for previously unseen pairs, an that the LC-UNC setup is usually performing as well as or better than alternative selection criteria in the one-query per data point setting.

## C.2 HUMAN EVALUATION

Tables C.3 and C.4 show additional results on the active learning from human feedback. As above, we tested our approach using different filtering thresholds $t$ on the two test sets $T$ (Table C.3) and $S$ (Table C.4). In the Retrain condition, the classifier $\hat{\varphi}$ was trained for a single epoch on all labeled datapoints $\bigcup_{i<n} D_i$ in order to combat potential issues with catastrophic forgetting. In the Retrain + Reweigh condition, the same was done, but the Cross Entropy loss was reweighed to balance the empirical label frequencies in $\bigcup_{i<n} D_i$. In the From Scratch setting, we train a new classifier on $\bigcup_{i<n} D_i$ for 5 epochs from scratch without first training it separately on any $D_i$. Again, datapoints are reweighed according to their empirical frequency in $\bigcup_{i<n} D_i$ in the From Scratch + Reweigh setting.

| Method/Dataset | $\varphi_2$ (Same category) | $\varphi_1$ (Word replacement) |
|---|---|---|
| Random sampling, 1 query | $75.1 \pm 3.6$ | $74.8 \pm 1.8$ |
| Random sampling, 3 queries | $71.6 \pm 3.9$ | $72.5 \pm 1.5$ |
| Random sampling, 5 queries | $70.7 \pm 2.7$ | $73.4 \pm 1.8$ |
| BALD 1 query | $75.9 \pm 4.0$ | $77.9 \pm 2.1$ |
| BALD 3 queries | $73.8 \pm 6.5$ | $78.1 \pm 1.7$ |
| BALD automatic relabeling | $76.1 \pm 4.5$ | $77.6 \pm 2.6$ |
| LC 1 query | $79.1 \pm 4.4$ | $78.5 \pm 1.8$ |
| LC 3 queries | $74.6 \pm 2.4$ | $79.5 \pm 1.8$ |
| LC automatic relabeling | $73.4 \pm 5.9$ | $78.2 \pm 1.3$ |
| LC-UNC 1 query | $79.0 \pm 4.9$ | $79.7 \pm 1.5$ |
| LC-UNC 3 queries | $75.8 \pm 5.4$ | $78.7 \pm 2.6$ |
| LC-UNC automatic relabeling | $76.6 \pm 3.9$ | $76.7 \pm 1.5$ |
| VARRA 1 query | $77.3 \pm 7.4$ | $78.9 \pm 2.1$ |
| VARRA 3 queries | $73.1 \pm 5.7$ | $79.8 \pm 1.6$ |
| VARRA automatic relabeling | $77.7 \pm 2.9$ | $78.0 \pm 1.3$ |
| Majority 1 query | $74.9 \pm 3.5$ | $76.8 \pm 2.4$ |
| Majority 3 queries | $78.7 \pm 5.2$ | $79.6 \pm 0.9$ |
| Majority automatic relabeling | $74.4 \pm 6.2$ | $77.9 \pm 1.8$ |

Table C.2: Balanced accuracy for BERT classifier trained using a constant amount of 50k gradient steps and a constant amount of 10k queries. All results are averaged over 10 runs and $\pm$ indicates the difference from the upper/lower bound of a naive $95\%$ confidence interval assuming normally distributed errors.

| Method | ACC | TNR | TPR |
|---|---|---|---|
| Baseline: Constant 0 | 78.8 | 100.0 | 0.0 |
| AL t=0.5 | $79.8 \pm 0.3$ | $97.2 \pm 0.3$ | $15.1 \pm 1.2$ |
| AL + Relabel t=0.5 | $81.1 \pm 0.3$ | $95.5 \pm 0.7$ | $28.6 \pm 2.2$ |
| AL + Relabel + Retrain t=0.5 | $79.6 \pm 0.4$ | $95.3 \pm 1.4$ | $21.5 \pm 3.9$ |
| AL + Relabel + Retrain + Reweigh t=0.5 | $79.6 \pm 0.8$ | $93.9 \pm 1.6$ | $26.6 \pm 3.4$ |
| From Scratch t=0.5 | $77.5 \pm 1.3$ | $90.8 \pm 3.3$ | $28.1 \pm 7.1$ |
| From Scratch + Reweigh t=0.5 | $77.7 \pm 1.4$ | $91.0 \pm 2.7$ | $28.3 \pm 5.0$ |
| AL t=0.1 | $80.0 \pm 0.5$ | $95.2 \pm 0.7$ | $23.7 \pm 3.5$ |
| AL + Relabel t=0.1 | $80.7 \pm 0.6$ | $93.0 \pm 0.9$ | $35.0 \pm 1.3$ |
| AL + Relabel + Retrain t=0.1 | $62.1 \pm 5.6$ | $61.5 \pm 8.9$ | $64.0 \pm 7.0$ |
| AL + Relabeling + Retrain + Reweigh t=0.1 | $52.8 \pm 6.2$ | $46.8 \pm 7.7$ | $75.0 \pm 4.6$ |
| From Scratch t=0.1 | $53.4 \pm 7.9$ | $48.6 \pm 14.3$ | $71.1 \pm 9.2$ |
| From Scratch + Reweighed t=0.1 | $54.8 \pm 6.7$ | $51.2 \pm 10.5$ | $67.9 \pm 9.1$ |
| AL t=0.01 | $78.7 \pm 1.1$ | $87.5 \pm 2.1$ | $45, 7 \pm 1.8$ |
| AL + Relabel t=0.01 | $78.3 \pm 0.7$ | $86.8 \pm 1.5$ | $46.6 \pm 2.5$ |
| AL + Relabel + Retrain t=0.01 | $21.2 \pm 0.1$ | $0.0 \pm 0.0$ | $100 \pm 0.0$ |
| AL + Relabel + Retrain + Reweigh t=0.01 | $21.1 \pm 0.0$ | $0.0 \pm 0.0$ | $100 \pm 0.0$ |
| From Scratch t=0.01 | $21.7 \pm 0.5$ | $0.0 \pm 0.0$ | $99.5 \pm 0.6$ |
| From Scratch + Reweigh t=0.01 | $21.8 \pm 1.5$ | $1.5 \pm 3.6$ | $98.3 \pm 1.7$ |

Table C.3: Results for active learning to predict human fairness judgments, on test data $T$. Active learning classifiers are retrained 10 times on the last batch $D_6$. Results are averaged and $\pm$ indicates the difference from the upper/lower bound of a naive $95\%$ confidence interval assuming normally distributed errors.

| Method | ACC | TNR | TPR |
|---|---|---|---|
| Baseline: Constant 0 | 96.1 | 100.0 | 0.0 |
| AL t=0.5 | $93.8 \pm 0.5$ | $97.0 \pm 0.6$ | $14.6 \pm 2.2$ |
| AL + Relabel t=0.5 | $92.1 \pm 0.6$ | $95.1 \pm 0.7$ | $18.9 \pm 2.7$ |
| AL + Relabel + Retrain t=0.5 | $90.7 \pm 1.7$ | $93.8 \pm 1.9$ | $12.8 \pm 4.0$ |
| AL + Relabel + Retrain + Reweigh t=0.5 | $89.0 \pm 1.3$ | $92.0 \pm 1.4$ | $16.4 \pm 3.4$ |
| From Scratch t=0.5 | $89.2 \pm 2.6$ | $91.8 \pm 2.5$ | $25.7 \pm 5.5$ |
| From Scratch + Reweigh t=0.5 | $89.2 \pm 2.5$ | $91.8 \pm 2.7$ | $25.7 \pm 4.4$ |
| AL t=0.1 | $90.4 \pm 1.3$ | $93.3 \pm 1.3$ | $21.0 \pm 2.3$ |
| AL + Relabel t=0.1 | $89.6 \pm 0.8$ | $92.2 \pm 0.8$ | $24.6 \pm 1.4$ |
| AL + Relabel + Retrain t=0.1 | $60.0 \pm 8.1$ | $59.5 \pm 8.8$ | $72.8 \pm 11.9$ |
| AL + Relabel + Retrain + Reweigh t=0.1 | $46.7 \pm 7.4$ | $45.2 \pm 8.0$ | $83.9 \pm 7.6$ |
| From Scratch t=0.1 | $50.6 \pm 10.4$ | $49.8 \pm 11.2$ | $69.6 \pm 9.3$ |
| From Scratch + Reweigh t=0.1 | $55.0 \pm 9.4$ | $54.5 \pm 10.0$ | $66.7 \pm 6.6$ |
| AL t=0.01 | $80.6 \pm 2.3$ | $82.3 \pm 2.7$ | $38.2 \pm 6.8$ |
| AL + Relabel t=0.01 | $80.2 \pm 1.3$ | $85.5 \pm 1.4$ | $30.0 \pm 2.7$ |
| AL + Relabel + Retrain t=0.01 | $3.9 \pm 0.0$ | $0.0 \pm 0.0$ | $100.0 \pm 0.0$ |
| AL + Relabel + Retrain + Reweigh t=0.01 | $3.9 \pm 0.0$ | $0.0 \pm 0.0$ | $100.0 \pm 0.0$ |
| From Scratch t=0.01 | $4.6 \pm 0.9$ | $0.0 \pm 0.1$ | $99.6 \pm 0.4$ |
| From Scratch + Reweigh t=0.01 | $5.4 \pm 3.9$ | $1.6 \pm 3.2$ | $50.8 \pm 1.6$ |

Table C.4: Results for active learning to predict human fairness judgments, using the separate test data $S$. Active learning classifiers are retrained 10 times on the last batch $D_6$. Results are averaged and $\pm$ indicates the difference from the upper/lower bound of a naive $95\%$ confidence interval assuming normally distributed errors.

# D    FURTHER DETAILS ON TRAINING DOWNSTREAM CLASSIFIERS

The downstream classifier $f$ consists of a pretrained roberta-base model followed by a linear layer with 768 neurons applied to the output embedding of the first token, a Tanh layer, another linear layer mapping to a single dimension, and a Sigmoid layer. We train $f$ using binary Cross Entropy reweighed to balance the empirical label frequencies in $D$ for 3 epochs using a batch size of 32 and the Adam optimizer with a learning rate of 0.00001.

Table D.1 extends Table 1 and shows that censoring words yields very strong constraint adherence for the respective word list [11]. However, we find it to generalize worse than CLP trained with the same word list, both to our style transfer pairs, and even to the respective other word list. Similarly, we find that training with CLP on $C^e$ or our style transfer pairs $C_3$ does not just yield significantly improved constraint adherence on $C_3$, but also generalizes better to $C_1$ and $C_2$ than the respective other of the two word replacement constraint sets without losing much downstream accuracy. Lastly, the table also shows that the better generalization from style transfer to word replacement persists for large values of $\lambda$ in CLP and that these values can provide strong improvements in terms of fairness, albeit at a larger cost in terms of balanced accuracy.

| Training/Evaluation | BA | $WR_{50}$ ($C_1$) | WR ($C_2$) | ST ($C_3$) | Full $C^e$ |
|---|---|---|---|---|---|
| Baseline | $88.4 \pm 0.1$ | $78.4 \pm 1.4$ | $81.3 \pm 1.5$ | $76.7 \pm 1.8$ | $78.5 \pm 1.5$ |
| Censoring $WR_{50}$ | $87.0 \pm 0.3$ | $99.8 \pm 0.0$ | $88.4 \pm 1.2$ | $84.7 \pm 1.1$ | $85.9 \pm 1.2$ |
| Censoring WR | $86.1 \pm 0.4$ | $91.4 \pm 1.2$ | $99.3 \pm 0.2$ | $89.0 \pm 1.5$ | $92.8 \pm 1.0$ |
| Censoring Both WR | $86.2 \pm 0.3$ | $99.7 \pm 0.2$ | $99.1 \pm 0.1$ | $89.3 \pm 0.4$ | $92.8 \pm 0.3$ |
| CLP($\lambda =5$) $WR_{50}$($C_1$) | $87.0 \pm 0.3$ | $98.3 \pm 0.1$ | $89.1 \pm 1.9$ | $86.3 \pm 1.9$ | $87.3 \pm 1.8$ |
| CLP($\lambda =5$) WR ($C_2$) | $87.2 \pm 0.1$ | $93.1 \pm 1.2$ | $98.2 \pm 0.4$ | $90.5 \pm 1.7$ | $92.9 \pm 1.2$ |
| CLP($\lambda =5$) ST ($C_3$) | $85.9 \pm 0.1$ | $95.3 \pm 0.4$ | $97.1 \pm 0.3$ | $95.4 \pm 0.4$ | $95.5 \pm 0.3$ |
| CLP($\lambda =5$) Full $C^e$ | $85.0 \pm 3.4$ | $95.5 \pm 0.9$ | $97.8 \pm 0.6$ | $94.9 \pm 0.9$ | $95.7 \pm 0.8$ |
| CLP($\lambda =125$) $WR_{50}$($C_1$) | $82.5 \pm 1.3$ | $98.3 \pm 0.6$ | $94.3 \pm 0.8$ | $90.9 \pm 1.1$ | $92.1 \pm 0.9$ |
| CLP($\lambda =125$) WR ($C_2$) | $81.8 \pm 1.5$ | $95.9 \pm 2.2$ | $98.6 \pm 0.5$ | $92.5 \pm 2.2$ | $94.7 \pm 1.5$ |
| CLP($\lambda =125$) ST ($C_3$) | $80.3 \pm 2.8$ | $97.6 \pm 0.8$ | $98.4 \pm 0.6$ | $97.2 \pm 0.9$ | $97.2 \pm 0.9$ |
| CLP($\lambda =125$) Full $C^e$ | $79.3 \pm 6.1$ | $97.8 \pm 1.3$ | $98.6 \pm 0.9$ | $97.1 \pm 1.6$ | $97.4 \pm 1.4$ |

Table D.1: Balanced accuracy and individual fairness (proportion of similar pairs $(s, s') \in C_i$ for which $f(s) = f(s')$) for a Roberta-based classifier $f$ trained with CLP using different constraint sets for training. Results reported with $\pm$ are averaged over 5 runs and $\pm$ indicates the difference from the upper/lower bound of a naive 95% confidence interval assuming normally distributed errors.

## D.1    EXPERIMENTS WITH FILTERING ON SYNTHETIC DATA

The filtering process for CLP is implemented as follows: for each batch $B$ of labeled training examples $(s, y(s))$ used to train a downstream classifier $f$, we evaluate $p_{\hat{\varphi}}(s, s')$ for all $(s, s') \in C^e$ with $s \in B$. Then, for every $s \in B$ we randomly select a pair $(s, s')$ among the pairs with $p_{\hat{\varphi}}(s, s') > t$ for a filtering threshold $t$ to use in the CLP regularizer $\lambda ||l(s) - l(s')||_2$ with $l$ representing the logits of the downstream classifier $f$, using $(s, s)$ if no such pair exists. To allow for more precise control over the statistical properties of $\hat{\varphi}$, we constructed additional $\hat{\varphi}$ using a look-up table using $\varphi_i$ and flip the labels of randomly selected pairs $(s, s')$ with either $\varphi_i(s, s') = 1$ or $\varphi_i(s, s') = 0$ in order to achieve specific True positive rates (TPR) and true negative rates (TNR). Table D.2 shows that there are consistent benefits from filtering for the synthetic similarity function $\varphi_1$ from App. C across different values of $\lambda$, even when an imperfect $\hat{\varphi}$ with a TPR and TNR of 75% is used.

D.3 shows that unlike for $\varphi_1$ (Table D.2), there is little gain from filtering constraints for $\varphi_2$, most likely because some of the constraint candidates generated by GPT-3 and our style transfer approach are difficult to enforce while maintaining high level of accuracy. While all of these constraints are inactive for $\varphi_1$ and are therefore not enforced with sufficiently accurate filtering, many of them remain active with $\varphi_2$ such that filtering yields no clear benefits.

---

[11]Artifacts like word replacement lists that contain both a word $s$ and substrings of $s$ keep this below 100%

| Method | Balanced Accuracy | Fairness |
|---|---|---|
| Baseline | $88.2 \pm 0.4$ | $82.0 \pm 2.2$ |
| Full $C^e$ $\lambda = 5.0$ | $85.6 \pm 0.4$ | $98.2 \pm 0.2$ |
| Full $C^e$ $\lambda = 125.0$ | $73.9 \pm 16.8$ | $98.6 \pm 1.0$ |
| Filtering with 75% TNR/TPR, $\lambda = 5.0$ | $86.3 \pm 0.6$ | $97.9 \pm 0.3$ |
| Filtering with 75% TNR/TPR, $\lambda = 125.0$ | $77.2 \pm 6.1$ | $99.1 \pm 0.3$ |
| Perfect filtering $\lambda = 5.0$ | $\mathbf{87.5} \pm 0.1$ | $98.2 \pm 0.2$ |
| Perfect filtering $\lambda = 125.0$ | $86.1 \pm 0.4$ | $\mathbf{99.3} \pm 0.1$ |

Table D.2: Balanced accuracy and individual fairness (proportion of similar pairs $(s, s')$ according to $\varphi_1$ for which $f(s) = f(s')$) CLP training after filtering $C^e$ using approximations of $\varphi_1$ with varying error profiles. All results are averaged over 5 runs and $\pm$ indicates the difference from the upper/lower bound of a naive 95% confidence interval assuming normally distributed errors.

| Method | Balanced Accuracy | Fairness |
|---|---|---|
| Baseline | $87.9 \pm 1.2$ | $76.5 \pm 1.5$ |
| Full $C^e$ $\lambda = 5.0$ | $85.6 \pm 0.4$ | $96.6 \pm 0.4$ |
| Full $C^e$ $\lambda = 125.0$ | $78.9 \pm 2.7$ | $\mathbf{97.5} \pm 1.2$ |
| Perfect filtering $\lambda = 5.0$ | $\mathbf{86.6} \pm 0.3$ | $95.7 \pm 0.6$ |
| Perfect filtering $\lambda = 125.0$ | $80.7 \pm 2.2$ | $97.3 \pm 0.6$ |

Table D.3: Balanced accuracy and indvidual fairness (proportion of similar pairs $(s, s')$ according to $\varphi_2$ for which $f(s) = f(s')$) for CLP training after filtering $C^e$ using approximations of $\varphi_2$ with varying error profiles. All results are averaged over 5 runs and $\pm$ indicates the difference from the upper/lower bound of a naive 95% confidence interval assuming normally distributed errors.

## D.2 EXPERIMENTS WITH FILTERING ON HUMAN FAIRNESS JUDGMENTS

Table D.4 is an extended version of Table 4 including additional experiments with a larger regularization parameter $\lambda = 125$. Again, there is no visible benefit from filtering. Counterintuitively, more filtering appears to correspond to less accuracy but slightly more fairness, but this might be by chance, given the significantly larger error bars for $\lambda = 125$.

| Method | BA | NR | Fairness ($T$) | Fairness ($S$) |
|---|---|---|---|---|
| Baseline | $88.2 \pm 0.4$ | $0.0$ | $82.1 \pm 2.1$ | $84.7 \pm 1.3$ |
| WR (Garg) $\lambda = 5$ | $87.1 \pm 2.0$ | $100$ | $92.8 \pm 0.9$ | $95.2 \pm 0.8$ |
| WR $\lambda = 5$ | $87.2 \pm 0.2$ | $100$ | $95.8 \pm 0.9$ | $95.8 \pm 1.2$ |
| Full constraint set $C^e$ $\lambda = 5$ | $85.9 \pm 0.3$ | $100$ | $96.5 \pm 1.4$ | $97.0 \pm 1.5$ |
| Filtering with threshold 0.5 $\lambda = 5$ | $85.9 \pm 0.5$ | $88.5 \pm 1.0$ | $97.4 \pm 1.1$ | $97.1 \pm 1.1$ |
| Filtering with threshold 0.1 $\lambda = 5$ | $86.1 \pm 0.1$ | $84.6 \pm 1.4$ | $97.2 \pm 0.6$ | $96.6 \pm 0.6$ |
| Filtering with threshold 0.01 $\lambda = 5$ | $85.9 \pm 0.2$ | $76.9 \pm 2.0$ | $97.1 \pm 1.0$ | $96.9 \pm 1.1$ |
| WR (Garg) $\lambda = 125$ | $81.6 \pm 0.6$ | $100$ | $95.6 \pm 1.7$ | $96.8 \pm 0.2$ |
| WR $\lambda = 125$ | $81.2 \pm 2.7$ | $100$ | $97.4 \pm 2.5$ | $97.5 \pm 0.1$ |
| Full constraint set $C^e$ $\lambda = 125$ | $81.8 \pm 2.1$ | $100$ | $98.0 \pm 0.6$ | $97.6 \pm 1.1$ |
| Filtering with threshold 0.5 $\lambda = 125$ | $81.3 \pm 1.5$ | $88.5 \pm 1.0$ | $98.1 \pm 0.9$ | $98.0 \pm 0.7$ |
| Filtering with threshold 0.1 $\lambda = 125$ | $80.7 \pm 1.9$ | $84.6 \pm 1.4$ | $97.8 \pm 1.3$ | $98.2 \pm 0.6$ |
| Filtering with threshold 0.01 $\lambda = 125$ | $80.7 \pm 2.2$ | $76.9 \pm 2.0$ | $98.6 \pm 0.5$ | $97.8 \pm 0.9$ |

Table D.4: Balanced accuracy, percent of unfiltered pairs, and individual fairness on two different test sets for differently trained classifiers $f$. Fairness is measured as the fraction of pairs $(s, s')$ with $\varphi(s, s') = 0$ according to the majority of annotators, for which $f(s) = f(s')$. Active learning classifiers are retrained 5 times on the last batch $D_6$ and results are averaged over 5 training runs. $\pm$ indicates the difference from the upper/lower bound of a naive $95\%$ confidence interval assuming normally distributed errors.

| Demographic Group | Descriptors | Nouns |
|---|---|---|
| Male | male,manly,masculine | man, men, grandfather, bro, guy, boy, father, dad, son, husbands, husband, grandpa, brother |
| Female | female, pregnant, feminine, femme, womanly | woman, women, grandmother, lady, ladies, girl, mother, mom, daughter, wives, wife, grandma, sister |
| Transgender | transsexual, FTM, F2M, MTF, trans, M2F, transgender, trans female, trans feminine, trans femme,trans fem, trans male,trans masculine,trans masc | |
| Heterosexual | straight, heterosexual, hetero | |
| Homosexual | gay, homosexual, lesbian | |
| Christian | Amish, Catholic, Christian, Evangelical, Lutheran, Mormon, Protestant, Unitarian | |
| Jewish | Conservative Jewish, Orthodox Jewish, Reform Jewish, Hasidic, Jewish | |
| Muslim | Muslim, Shia, Sunni | |
| Hindu | Hindu | |
| Buddhist | Buddhist | |
| Atheist | atheist, irreligious | |
| Black | African-American, African, Black, BIPOC | |
| White | European-American, European, Caucasian, White | |
| Asian | Chinese-American, American-Born Chinese, Chinese, Indian-American, Indian, Korean-American, Korean, Vietnamese-American, Vietnamese, AAPI | |
| Latino | Cuban-American, Cuban, Dominican-American, Dominican, Salvadoran-American, Salvadoran, Guatemalan-American, Guatemalan, Mexican-American, Mexican, Filipina-American, Filipina, Filipino-American, Filipino, Hispanic, Latinx, Latine, Latino, Latina, Latin American | |

Table E.1: Lists of words used for word replacementm, based on Smith et al. (2022)

## E    WORD LISTS AND EXAMPLE GENERATIONS

Tables E.2–E.4 show 5 randomly selected example pairs $(s, s')$ produced by our style transfer approach and GPT-3 in zero-shot and edit mode. Table E.5 shows 5 generated comments for each method using the same source comment and target demographic attribute.

Warning: Some of the example texts contain offensive language.

| Original Comment | Modified Comment |
|---|---|
| 'Hahaha, really? Thousands of years of war and it was mostly white men, huh? Read some history about non-European cultures to pierce your idyllic bubble.' | 'Hahaha, really? Thousands of years of war and it was mostly white European Christians, huh? Read some history about non-European cultures to pierce your idyllic bubble' |
| 'I would really like to see your source for the Japanese fear of the Russians as being any kind of factor whatsoever. For most of the Japanese people, the Russians were not even on the radar screen!!' | 'I would really like to see your source for the Buddhist fear of the west as being any kind of factor whatsoever. For most of the western people, the west and religion were not even on the radar screen!!' |
| 'Yes: at least implicitly it endorses the old canard that the United States is a "Christian nation."' | 'Yes: at least implicitly it endorses the old canard that the United States is a "white racist nation."' |
| 'Please be careful what you label fake news. News of shootings (Greenville?), migrants in Sweden raising crime rates, Hispanics being rapists and murderers, illegal voting, etc those are examples of fake news....feet on couch? not so much....' | 'Please be careful what you label fake news. Hundreds of shootings (Green Mountain?), migrants in Sweden raising crime rates, homosexuals being raped and murderers, illegal voting, etc those are examples of fake news....feet on couch? not so much.' |
| 'And often they are not so silent. Gay friends have told me about various such clergy who are well known – only no one speaks out about it.' | 'And often they are not so silent. Gay friends have told me about various such men who are well known – only no one speaks out about it.' |

Table E.2: Examples of pairs produced by our style transfer approach

| Original Comment | Modified Comment |
|---|---|
| 'OK, so now Iḿ confused. Is it or is it not OK to mock a reporterś physical handicap? Call a Latina beauty queen "Miss Housekeeping"? Post a mock video showing yourself clocking a woman you dont́ like with a hard-hit golf ball?' | ' Is it or is it not OK to mock a reporterś physical handicap? Call a Muslim beauty queen "Miss Housekeeping"? Post a mock video showing yourself clocking a woman you dont́ like with a hard-hit golf ball?' |
| "You caught me. I confess. I've been lying this whole time. I'm not actually an atheist, of course I believe in a god. I see him in the mirror every day !" | " You caught me. I confess. I've been lying this whole time. I'm not actually a homosexual, of course I believe in a god. I see him in the mirror every day !" |
| "Denying the existence of racism in America,(or most anywhere), is just ignoring reality. The formal dicrimination has largely been curbed(with exception of using race to design Congressional districts), but those wonderful 'family values' of the white South still carry the mythology of inherent white superiority" | ". Denying the existence of racism in America,(or most anywhere), is just ignoring reality. The formal dicrimination has largely been curbed(with exception of using race to design Congressional districts), but those wonderful 'family values' of the Hindu South still carry the mythology of inherent Hindu" |
| 'Cite to "the Republicans, Trump being a Republican, who are against gays" please. New one to me. (Other than Trump being registered Republican of course, lol!)' | ' Christian people are against gays.' |
| "You'd better watch out if you don't claim Christianity unto yourself like they do. It is true that people harbor resentment over people whom they view as different. It's the only reason I can think of why they hate gay people." | " You'd better watch out if you don't claim Christianity unto yourself like they do. It is true that people harbor resentment over people whom they view as different. It's the only reason I can think of why they hate Asians." |

Table E.3: Examples of pairs produced by GPT-3 (zero-shot)

| Original Comment | Modified Comment |
|---|---|
| 'So do they cover their faces? This is about being able to identify women for practical purposes. Makes sense to moi.' | 'So do they cover their faces? This is about being able to identify White people for practical purposes. Makes sense to moi. ' |
| 'What about LGBTQI+M(xn) W(omxn)Y(oxth)C(hxldrxn)O(thxrs)' | 'Christianity is the best religion. Christian people are the best people. Jesus Christ was a man who lived a perfect life and died for our sins. We should all be following Jesus because he is the author and finisher of our faith. ' |
| 'Bust Judaism, Christianity and Islam are all Abrahamic religions tracing their origins to the patriarch Abraham. They are all related.' | 'Bust Hispanic people are all the same. They are all related. ' |
| "AO96 doesn't allow men into womens rest rooms or locker rooms. You do realize that trans people have been using the bathroom with you forever right? Have you ever been injured by any of these people in all the years you have been using public facilities?" | "AO96 doesn't allow white people into black rest rooms or locker rooms. You do realize that trans people have been using the bathroom with you forever right? Have you ever been injured by any of these people in all the years you have been using public facilities? " |
| 'so.. I . have seen dozens of women in niqab in Toronto this week alone Almost every day I see groups of them. The problem with the niqab policy is that is was for such a silly narrow issue. The niaqb should have been banned from public buildings and shopping malls.' | 'so.. I . have seen dozens of black people in the niqab in Toronto this week alone Almost every day I see groups of them. The problem with the niqab policy is that is was for such a silly narrow issue. The niaqb should have been banned from public buildings and shopping malls. ' |

Table E.4: Examples of pairs produced by GPT-3 (edit mode)

| Original Comment | Modified Comment (Style Transfer) | Modified Comment (GPT-3 zero shot | Modified Comment (GPT-3 edit mode) |
|---|---|---|---|
| I agree with those many trans-gendered. | I agree with those many pre-gospel Catholics. | I agree with those many Christians. | I agree with those many Christian people. |
| Everyone in plain black pajamas might do the trick. | Everyone in plain white pajamas might do the trick | All white people in plain white pajamas might do the trick. | Everyone in plain white pajamas might do the trick. |
| You do know Mexicans are American ...right? | You do know gays are gay...right? | You do know homosexuals are American ...right? | You do know Homosexuals are people too ...right? . |
| I'm an atheist and I don't care about obnoxious atheists. Either way. | I'm an Indian and I don't care about obnoxious Muslims. Either way. | I don't care about obnoxious Hindus. | I'm a Hindu and I don't care about obnoxious Hindus. Either way. |
| Another condescending ignorant Hindu nationalist ignorant of the genocidal, racist and murderous past of its culture. | Another condescending ignorant white nationalist ignorant of the genocidal, racist and murderous past of its culture | Another condescending ignorant white nationalist ignorant of the genocidal, racist and murderous past of its culture. | A White person. |

Table E.5: Examples of pairs produced by different methods for the same source comment and target demographic attribute.

# F  GROUP FAIRNESS

Table F.1 provides results on equality of odds for a subset of the models trained for the experiment presented in Table D.4. Specifically, for every group $j$ that was considered for generation, we calculated the TPR and TNR on the test set restricted to comments $s$ mentioning that group (label $y_j(s)$=1). We then calculated the absolute pairwise differences between the TPRs/TNRs for every pair of groups (in percent), and present the mean and maximum over these for each model. We find that CLP-training improves the TNR gap at the cost of a worse TPR gap, except for pairs generated by the Word Replacement list of Garg et al. (2019). This effect is most extreme when using our full constraint set $C^e$ and more mellow for Word Replacement based on the word list of Smith et al. (2022) or for the filtered $\hat{C}^r$. The improved TNR gap at the cost of a worse TPR gap is directionally consistent with the results reported by Garg et al. (2019) for CLP training using their word list, with groups defined by the presence of specific identity terms from the word list rather than the labels $y_j(s)$. We echo Garg et al. (2019)'s recommendation for practicioners to select a method depending on their relative prioritization between improving individual fairness and equitable True Negative Rates, compared to equitable True Positive Rates.

| Method | TPR Gap (Mean) | TNR Gap (Mean) | TPR Gap (Max) | TNR Gap (Max) |
|---|---|---|---|---|
| Baseline | 5.8 | 20.9 | 13.6 | 54.3 |
| WR (Garg) $\lambda = 5$ | 5.4 | 20.3 | 13.6 | 51.8 |
| WR $\lambda = 5$ | 11.2 | 10.8 | 34.7 | 30.1 |
| Full constraint set $C^e$ $\lambda = 5$ | 15.7 | 6.1 | 56.3 | 22.0 |
| Filtering with $t = 0.5$ $\lambda = 5$ | 11.1 | 10.0 | 34.5 | 29.3 |

Table F.1: Mean and Maximum of absolute pairwise TPR and TNR gaps (Absolute differences between TPR, in percent) between comments mentioning different demographic groups for differently trained classifiers $f$.

