# OpenReview forum: "Human-Guided Fair Classification for Natural Language Processing"
_ICLR.cc/2023/Conference — ICLR 2023 notable top 25%_

### Official Review · Reviewer_cacL · 2022-10-24

**Confidence:** 4
**Correctness:** 3
**Technical Novelty And Significance:** 3
**Empirical Novelty And Significance:** 3
**Recommendation:** 8

**Clarity, Quality, Novelty And Reproducibility:**

In this section, I will provide more details of my comments.
First of all, I think the paper is technically sound overall. The tasks and paper considered in the paper are a novel combination of well-known techniques. In particular, the paper bridge the gap between human intuition about input perturbations for individual fairness and the formal similarity specifications capturing them.

However, I think the paper could be further improved in terms of clarity and discussion of multiple related notions (e.g., group fairness) and the potential broader impacts.

While the paper is fairly readable, there is substantial room for improvement in clarity. The techniques involved in this paper span multiple domains, requiring the reader to have a good knowledge of algorithmic fairness, NLP, user study, etc. I have seen the authors put good effort into giving details in the main text and appendix. However, it is hard to understand the experiment sections without referring to the appendix in detail. For example, in Table 2, the authors do not provide any description of the "Metric/Method" column in the main text. Similar problems happen in Table 3. I also have a question about evaluating the diversity of generated pairs: the paper uses the fairness metric as the evaluation criterion for models trained on different constraint sets. The word choice of "diversity" sounds strange, given the evaluation criterion. A better word choice might be something like "coverage".

Since the paper studies fairness. I am also interested in how the generated intuitive fairness specification affects group fairness notions such as demographic parity and equalized odds. Besides, since the fairness specification is intuitively generated by humans, how do you provide more documentary specifications on this to the relevant stakeholders, such as ML/NLP practitioners and model users, to trust the overall workflow?

Lastly, I have some questions regarding reproducibility: I only see the codes in supplementary materials. Will the dataset be released? If yes, how do you plan to release them in order to provide a better potential impact in the long run?


**Strength And Weaknesses:**

Strengths:
1. The paper considers a novel approach to bridge the gap between human intuition about input perturbations for individual fairness and the formal similarity specifications capturing them.
2. The methods to diversify expressive candidate pairs and active learning for quality control of the dataset are technically sound.

Weaknesses (see more details in the section below):
1. The paper needs further improvement in clarity.
2. Beyond individual fairness, a discussion of how the proposed dataset with individual fairness would affect group fairness notions is needed.
3. The broader impact of how relevant stakeholders would use the generated workflow is unclear.


**Summary Of The Paper:**

The paper presents a framework for generating a dataset of diverse candidate pairs for individual fairness specifications by a set of methods (e.g., an extended word replacement list, unsupervised style transfer, and zero-shot modification using GPT-3). To align with human fairness intuitions for a considered downstream task, the authors propose active learning approaches to improve the quality of the datasets. Empirical studies are conducted to show the proposed dataset is of high quality.

**Summary Of The Review:**

The paper is novel and technically sound but needs improvements on clarity, and discussion of broader impact, given that the paper studies fairness.

---

> ### Author Response · Authors · 2022-11-14
> **Response to Reviewer cacL**
>
> Dear Reviewer,
>
> Thank you for your insightful review. We are encouraged that you found our approach novel and technically sound. We have added additional details to our experiments section to improve clarity. We have also added experimental results on group fairness to our appendix. We provide more detailed answers to your questions, including a detailed discussion on documentary specification for stakeholders, below.
>
> **The clarity of the paper should be improved**
>
> Thank you for your suggestions for improving the clarity of our paper. In the revised version, we have reworked the experiments section in order to increase readability without the need to consult the appendix excessively. Specifically, we have added additional details about experimental results, both in the respective table captions and in the main text of the experiment section.
>
> **When evaluating the diversity of generated pairs, the word “coverage” might be more suitable than “diversity”.**
>
> We appreciate this suggestion and have replaced references to “diversity” with “coverage” where appropriate.
>
> **How do the generated intuitive fairness specifications affect group fairness notions such as demographic parity and equalized odds?**
>
> [Garg et al. (2019)](https://dl.acm.org/doi/pdf/10.1145/3306618.3317950) provide a discussion on the relationship between fairness specifications based on Word Replacement and Equality of odds, finding that both concepts are theoretically orthogonal. We added an additional experiment on equalized odds to App. F, finding similar results: compared to the baseline, CLP training on the full set $C$ of generated pairs reduces the mean TNR gap between groups from 20.9 to 6.1 but increases the mean TPR gap from 5.8 to 15.7. Using the filtered dataset  $\hat{C}^\star$ yields a mean TPR gap of 11.1 and a mean TNR gap at 10.0.
>
> We do not believe that demographic parity is a desirable goal in toxicity classification, as comments mentioning minorities are sadly often disproportionately likely to be toxic [(Dixon et al., 2018)](https://dl.acm.org/doi/10.1145/3278721.3278729), such that enforcing demographic parity according to mentions of demographic groups would lead to a toxicity detector either missing toxic comments insulting minorities it could have detected, or flagging additional harmless comments that are not about minorities.
>
> Part 1/2

---

> > ### Author Response · Authors · 2022-11-14
> > **Part 2**
> >
> > **Since the fairness specification is intuitively generated by humans, how do you provide more documentary specifications on this to the relevant stakeholders, such as ML/NLP practitioners and model users, to trust the overall workflow?**
> >
> > We consider multiple aspects to this question. First, we provide a detailed description of our human evaluation in App. A, and plan to release the fully anonymized dataset of human responses. We believe that combined with our experimental results, this enables practitioners to make an informed decision on whether to put trust in our framework for generating comment pairs and validating them based on human fairness judgments, as well as the specific pairs and human fairness judgments we collected for toxicity classification.
> >
> > Second, in our ethics statement we recommend that practitioners combine our framework with similarity labels provided by experts and stakeholders specific to the planned application whenever possible. This both ensures that the collected labels represent valid fairness judgments for that specific context, and helps build trust with stakeholders by explicitly involving them into the workflow. We have further expanded on this point in the ethics statement of the revised version of the paper.
> >
> > We believe that it is exceedingly difficult to capture human intuitions about individual fairness in complex domains like text using easily understandable rules. Because of this, we chose to focus on implicit specifications via human-labeled data, for which documentary specification beyond what is described is hard to provide. We see attempts at extracting more human-readable specifications from classifiers trained on these labels using techniques for explainable machine learning as an exciting direction for future research that could instill further trust in specifications produced by our framework. Until then, as for any other machine learning metric that is averaged over individual data points, the performance of our individual fairness metric on an unseen test set provides an unbiased estimate of the expected performance on random data drawn from the same distribution. This means that strong evidence about future performance in terms of individual fairness on human-validated pairs randomly generated by our method can be obtained from strong performance on a sufficiently large test set. Of course, individual fairness might only be violated on pairs that are missed by our generation approach, but we expect this to be less problematic given our improved coverage to previous approaches based on word replacement.
> >
> > **Will the dataset be released and how?**
> >
> > We plan to make our generated fully anonymized dataset (together with the crowdsourced labels) publicly available. A link to the dataset will be provided in the camera-ready version of this paper. When releasing the dataset, we will ensure that we follow the recommendations from [(Gebru et al., 2021)](https://cacm.acm.org/magazines/2021/12/256932-datasheets-for-datasets/fulltext) where appropriate and will include a datasheet for this dataset.
> >
> > [1] Gebru et al., Datasheets for datasets, Communications of the ACM 2021

---

> > > ### Comment · Reviewer_cacL · 2022-11-27
> > > **Response to Authors**
> > >
> > > Thanks for your detailed response. I will increase my score.

---

### Official Review · Reviewer_6aFV · 2022-10-24

**Confidence:** 4
**Correctness:** 3
**Technical Novelty And Significance:** 3
**Empirical Novelty And Significance:** 2
**Recommendation:** 8

**Clarity, Quality, Novelty And Reproducibility:**

**Clarity**:

The paper is written clearly, is well organized, and the approach is simple and straightforward.

**Quality**:

The paper is of good quality, but it would be better if additional experiments are added to investigate more various tasks.

**Novelty**:

Although the problem of word replacement has been previously investigated, the addition of style transfer and zero-shot capabilities is an interesting touch.

**Reproducibility**:

The Appendix has some info for reproducibility.

**Strength And Weaknesses:**

**Strengths:**

1. The paper studies interesting and timely problem.
2. The paper is written clearly and is well-organized.
3. The approach is simple and easy to follow.

**Weaknesses:**

1. It seems like the approach would be really task dependent. I think it is good for authors to acknowledge this somehow and mention how the task can affect the outcomes.
2. Related to my above comment, authors only study toxicity problem. It would be good to study other tasks and compare the differences and provide more in depth discussion on what the differences are in terms of the filtering approach per task as well as human alignment results.
3. While the approach is effective in achieving more "fair" outcomes, the approach imposes loss on accuracy.
4. Given that the task of toxicity detection is a subjective task and in many cases culturally dependent, what mechanisms authors take into account in their human studies other than recruiting more than one worker to label a hit and taking the maximum. Please provide some discussion on this.

**Additional Minor Comment:**

I would suggest authors refer to fairness through awareness instead of individual fairness as really what authors do is to study groups in their tasks (e.g., Muslim vs Christian) but using the notion of fairness through awareness, so to avoid this minor confusion, I would suggest changing the naming.

**Summary Of The Paper:**

The paper proposes a new framework to satisfy individual fairness notion by utilizing style transfer along with zero-shot techniques. They also perform crowdsourcing experiments to verify how much aligned results those techniques generate results compared to human intuition. They also use these human signals as means to improve they system for filtering pairs as well as training downstream models.

**Summary Of The Review:**

The paper studies an important problem and is written clearly; however, some additional experiments can benefit the paper. Including some discussions around my concerns raised in the weaknesses section.

---

> ### Author Response · Authors · 2022-11-14
> **Response to Reviewer 6aFV**
>
> Dear Reviewer,
>
> Thank you for your constructive review. We are pleased that you found our paper to be clear and of good quality. We agree that human intuitions about fairness are both subjective and task-dependent and have extended our discussion of these issues in our ethical considerations section. We provide detailed answers to your comments below.
>
> **The approach seems to be task dependent. It would be good if the authors acknowledge this and mention how the task can affect the outcomes. In addition, only toxicity classification was studied and it would be good to study other tasks and provide more in depth discussion on what the differences are in terms of the filtering per task and human alignment results.**
>
> We agree that the approach is task dependent, and in the original submission we tried to acknowledge this in the introduction. We state that “the similarity of inputs can be highly task dependent” and subsequently give an example where two inputs can be similar in the context of one task but not in another. We highlight that the central challenge we consider is the generation of a diverse set of similar inputs “in the context of a fixed text classification task” and that we instantiate the proposed pipeline “in the context of toxicity classification”. In our revised version we have added similar emphasis on the task-specific instantiation of our approach to the beginning of the methods and the experiments section.
>
> We think that there are two levels to the transferability of our approach to other tasks. First, our evaluations show that our approach performs strongly in terms of modifying demographic attributes while preserving content otherwise, and these results are not specific to toxicity classification (but could depend on the data domain of online comments). This indicates that our generation approach is likely suitable for other downstream tasks, at least in similar domains, for which invariance to mentions of different demographic groups is desired. Second, the agreement of human annotators with the fairness-relevance of generated pairs is clearly more task dependent. This is why in our ethical considerations section we advocate for basing filtering on fairness judgments provided by application-specific stakeholders when our framework is deployed by practitioners. We have added additional discussion to the ethical considerations section to further clarify this point.
>
> Lastly, we agree that testing our approach for other downstream tasks is an exciting direction for future work, but given the already broad scope of our work and the resources needed to study another task (e.g., payment to human annotators), we found it prudent to focus our human evaluation experiments on a single context.
>
> **While the approach is effective in achieving more "fair" outcomes, the approach imposes loss on accuracy.**
>
> In general, there appears to be an inherent trade-off between accuracy and fairness that is also commonly observed in other works on individual fairness ([Ruoss et al., 2020](https://proceedings.neurips.cc/paper/2020/file/55d491cf951b1b920900684d71419282-Paper.pdf); [Yurochkin & Sun, 2020](https://arxiv.org/abs/2006.14168); [Peychev et al., 2021](https://arxiv.org/abs/2111.13650)). While including pairs generated by Style Transfer and GPT-3 for CLP training slightly worsens this tradeoff, it substantially increases the coverage of fairness constraints. In particular, as seen in table D.4. our approach yields similar adherence to fairness constraints on a mix between Word Replacement, Style Transfer and GPT-3 pairs with a regularization strength of $\lambda=5$ (BA = 86%) as CLP training on Word Replacement pairs only with $\lambda=125$ (BA = 82%).
>
> Part 1/2

---

> > ### Author Response · Authors · 2022-11-14
> > **Part 2**
> >
> > **Toxicity detection is a subjective task and in many cases culturally dependent. What mechanisms were taken into account in the human studies (other than recruiting several workers to label a hit and taking the majority)?**
> >
> > We strongly agree with the importance of cultural dependence. As highlighted in the ethical considerations section, we encourage practitioners seeking to apply our framework to conduct small scale replications of our human study within their specific deployment context and involving relevant stakeholders to validate the suitability of our approach in that context. The correct approach to account for variation between workers ultimately comes down to context-specific considerations regarding the costs of false positives and false negatives as evaluated by different stakeholders.
> >
> > We thank the reviewer for raising this important point and have added additional details to the ethical considerations section, including a note on the need for periodic reassessment of collected fairness judgments to account for context shifts [(Aroyo et al., 2019)](https://dl.acm.org/doi/abs/10.1145/3308560.3317083) as well as suggestions for future work to incorporate distributional information about stakeholder responses by using Jury Learning [(Gordon et al., 2022)](https://dl.acm.org/doi/abs/10.1145/3491102.3502004) or multi-annotator-architectures [(Davanei et al., 2022)](https://direct.mit.edu/tacl/article/doi/10.1162/tacl_a_00449/109286/Dealing-with-Disagreements-Looking-Beyond-the).
> >
> > [1] Aroyo et al., Crowdsourcing Subjective Tasks: The Case Study of Understanding Toxicity in Online Discussions, WWW 2019
> >
> > [2] Gordon et al., Jury learning: Integrating dissenting voices into machine learning models, CHI 2022.
> >
> > [3] Davani et al., Dealing with disagreements: Looking beyond the majority vote in subjective annotations, Transactions of ACL 2022.
> >
> > **”Fairness through awareness” might be a more appropriate term than “individual fairness” as the paper studies groups in the toxicity classification task.**
> >
> > While we agree that the wording of “individual fairness” can be confusing as it refers to _individual comments_ rather than individual persons in our case, we use the term “individual fairness” to stay consistent with the nomenclature in prior work on similar data [(Yurochkin & Sun, 2020)](https://arxiv.org/abs/2006.14168). We have also considered the term “counterfactual fairness” used in [(Garg et al., 2019)](https://dl.acm.org/doi/pdf/10.1145/3306618.3317950), but decided to avoid it due to the contentiousness of the proper definition for counterfactuals in text.

---

> > > ### Comment · Reviewer_6aFV · 2022-12-03
> > > **Response to Authors**
> > >
> > > I really appreciate authors' time and extensive effort in providing answers to my concerns. I will also increase my score considering the author response.

---

### Official Review · Reviewer_Cvse · 2022-10-24

**Confidence:** 3
**Clarity, Quality, Novelty And Reproducibility:** This paper is fairly original, clear,…
**Correctness:** 4
**Technical Novelty And Significance:** 3
**Empirical Novelty And Significance:** 3
**Recommendation:** 8

**Strength And Weaknesses:**

The paper addresses several important issues in evaluation text classification models. The issue with evaluation datasets is that there is no evaluation data for certain demographics, let alone parallel data for individual fairness evaluation. Secondly, word replacement methods (which is often used in prior literature) does not check if the parallel data is indeed valid. Especially for nuanced topics like hate speech or t toxicity classification, swapping words is not enough, since a stereotype about one group may not exist for another group (e.g. older people are nurses is not as much of a stereotype as women being nurses in the US). The paper is generally strong in terms of rigor of methodology, novelty of method, and in contribution.

For weaknesses, the authors could be clearer on what fairness metric is being used for tables such as Table D.3. Additionally, an ablation study could be added in terms of how changing the similarity classification threshold affects performance of the similarity model (and the downstream performance of downstream classifiers)? Finally, I would have liked to see in the appendix table E1-4, the generations from the three methods on the same examples.

There are also a few nits, for example, make sure to reference all variables in graphs (for example, D in Figure 1).

As a note, the authors should reference this paper as well for automatic generation of evaluation data: https://arxiv.org/pdf/2202.03286.pdf.


**Summary Of The Paper:**

The problem this paper makes progress in solving is that of generating data to drive individual fairness constraints on text classification models, with a case study of toxicity classification. The authors make use of a data generation pipeline, such that, starting with an initial dataset of text with mentions of demographic information. First, they create an initial set of counterfactual examples using word replacement, GPT-3 (with three variants of sub-pipelines), and an unsupervised style transfer model. For the last two, they also train a BERT model to detect group presence in a text, which is used as an initial filter for GPT-3 generations and used to train the style transfer model. They then train a "similarity" model to detect, given an original text and its generated counterfactual, if both should be treated the same. This similarity model is trained using active learning. They then use this similarity model to iteratively filter out the initial set of pairs to a final set, which is used to train the downstream classifier.
Using toxicity classification, they show that classifiers trained on this method was able to improve on fairness, and showed that the generated pairs generally aligned with human annotations' intuition of fairness. They also did ablation studies on downstream model performance given each initial method of data creation, and the effect of active learning on the similarity model.

**Summary Of The Review:**

I think this is an important problem, and this work will help equip future researchers with another method to generate high-quality data, which can be used for both evaluation of text classification models or to train them.

---

> ### Author Response · Authors · 2022-11-14
> **Response to Reviewer Cvse**
>
> Dear Reviewer,
>
> Thank you for your kind review. We are excited that you appreciated the contribution and the novelty of our work. We respond to your comments below.
>
> **What is the definition of the fairness metric that is being used?**
>
> Throughout the paper, when evaluating the individual fairness of a classifier, we compute the percentage of pairs $(s, s’)$ in a test pool, which are classified the same by the classifier (i.e., $f(s)=f(s’)$). In our revised draft, we make the use of this metric, as well as the test pools used for evaluation, more explicit.
>
> **An ablation study could be added demonstrating how changing the similarity classification threshold affects performance of the similarity model (and the performance of the downstream classifier)?**
>
> We provide results for three different threshold levels in Table 3 in the main text and Tables C.3, C.4 and D.4 in the appendix.
>
> As expected, the TPR of the similarity classifier improves with smaller thresholds while the TNR becomes worse. However, this happens at a surprisingly slow rate with the TNR still at 87% at the threshold 0.01. In terms of the performance of the downstream toxicity classifier, we find very limited variation between the thresholds 0.5 and 0.01 with the error bars for individual fairness in D.4. mostly overlapping between the different thresholds.
>
> Please, let us know if there are any further experiments you would like to see in this direction.
>
> **I would like to see generations from the three methods on the same examples (Appendix tables E1-4).**
>
> We included additional examples of generations based on the different methods using the same source comment and target demographic attribute in Table E.5 in App. E.
>
> **Make sure to reference all variables in graphs (for example, $D$ in Figure 1).**
>
> Thank you for your careful reading of our paper and for pointing this out to us. We have fixed this, by labeling $D$ as the task data set in Figure 1. In general, we have tried to further improve the clarity and the readability in the updated version of our paper.
>
> **The authors should reference this paper for automatic generation of evaluation data: [Red Teaming Language Models with Language Models](https://arxiv.org/pdf/2202.03286.pdf).**
>
> Thank you for pointing out this work to us. We added the reference in the corresponding paragraph in related work.

---

### Official Review · Reviewer_Y9Ec · 2022-10-25

**Confidence:** 4
**Correctness:** 3
**Technical Novelty And Significance:** 3
**Empirical Novelty And Significance:** 2
**Recommendation:** 6

**Clarity, Quality, Novelty And Reproducibility:**

In general, I enjoyed reading the paper	and learning about the
proposed methodlogy. I had a hard time processing the multitude of
experimental results. In particular, it's not clear what the numbers
represent in each table. I would clearly describe in the caption of
each figure what the rows/columns represent and what metric is
used. In particular, the paper uses "fairness" without clearly
explaining how fairness is measured.

Some of	the design decisions are not well explained. In	particular, I
am curious about the maximum sequence size of 64 (which is quite
limiting). Is that to ensure that the style transfer and the GPT3
generation produce quality results?

Not clear what WR50 means.

In Table 1, all columns except for BA represent some measure of
fairness? If that's the case and if I understand the results WR (which
I'm guessing stands for word replacement) seems quite performant. How
do you justify the complexity of adding the style transfer and GPT3 on
top. What is the real advantage of the full C dataset?

The fairness metric used seemed a bit ad hoc. One of the more
standard metrics could be use (see Fairness Definitions Explained and
the following paper for an example of using equalized odds: Your
Fairness May Vary: Pretrained Language Model Fairness in Toxic Text
Classification - analysis of many LMs wrt fairness and model
size/training size/random seed (in the context of toxic text
prediction)).

Last but not least, I think "fairness specifications" is misleading
and quite an overblown term for what the paper is about; it made me
think of some theoretical formalism/specification. Similar pairs of
text across protected groups is a much more honest description of what
the paper generates.

**Strength And Weaknesses:**

Strengths:
- The methodlogy could be used in different contexts for generating pairs of similar senteces that differ only in the protected group
- Lots of experiments to support the methodology

Weakness:
- Some ad-hoc decisions (e.g., shortening the max sequence size) that are not well explained
- Lots of experimental results that are not well explained

**Summary Of The Paper:**

This paper introduces a workflow/methodology to generate pairs of
similar sentences that differ only wrt target/protected populations
such as gender or race. The methodology inclues increasingly
sophisticated steps, such as word replacement, unsupervised style
transfer and generation using GPT3. Crowdsourcing and active learning
is used to filter the generated pairs. The methodology is demonstrated
in the context of toxic text prediction. Using the pair-wise sentences
that vary only along the protected groups, a RoBERTa model is trained
by enforcing the logits for the two pieces of text to be similar
(since the toxic text prediction should not depend on the protected
subgroup used in the sentence).

**Summary Of The Review:**

This paper introduces a methodology to generate pairs of similar
sentences that differ only wrt protected group. The pairs generated
can be used to train fairer classifiers by imposing similar logits for
similar sentences.

---

> ### Author Response · Authors · 2022-11-14
> **Response to Reviewer Y9Ec**
>
> Dear Reviewer,
>
> Thank you for your thorough review. We are glad that you enjoyed reading our paper and see the potential for our method to be applied in additional contexts. We improved the clarity of our draft, discussed our individual fairness metric and explored the equality of odds in a group fairness experiment. Finally, we suggest an alternative title for the paper that might better capture the content, and would be grateful for your input on that. We provide detailed answers to your comments next.
>
> **It is not clear what the numbers and the rows/columns represent in each table. What metric is used and how is fairness measured?**
>
> Throughout the paper, we measure fairness in terms of the percentage of pairs $(s,s’) \in K$ for which the classifier produces equal outcomes (i.e, $f(s)=f(s’)$), where $K$ is a certain test pool of similar pairs (we informally define how we measure fairness in the last paragraph of Sec. 4.1 and refer to it as “individual fairness” in the revised paper to avoid confusion). In Table 1, the test pool is one of $C_1$, $C_2$, $C_3$, or $C$ – sets of pairs produced by different generation methods. In Table 4, the test pool is a subset of $T$. $T$ is a set of 500 pairs $(s, s’)$ randomly sampled from the test pool $C$ that combines all the different generation methods and in Table 4 we evaluate individual fairness on all pairs $(s,s’) \in T$ for which the majority of human annotators believed equal treatment is warranted (these are 78.8% of all pairs in T, as noted in Sec. 4.4).
>
> We have reworked the experiments section, moving some auxiliary results to the appendix and providing additional details on the experimental setups in the main text and table captions instead. We hope that these changes make our experimental section more informative and clear.
>
> **Why is the maximum sequence size set to 64? Is that to ensure the style transfer and the GPT3 generation produce quality results?**
>
> We limited the sequence size to 64 due to computational costs, and costs for human labeling. While 53.3% of the original dataset is shorter than 64 tokens, 5% are longer than 213 tokens. Accounting for such outliers would have slowed down our experiments and forced us to use substantially smaller batch sizes. Moreover, high variance in text length would have forced us to either substantially increase payments to crowdworkers, or accept that workers only receive adequate compensation on average but not necessarily for every individual task they complete.
>
> **What does WR50 mean?**
>
> WR50 refers to word replacement based on the 50-word list used in [(Garg et al., 2019)](https://dl.acm.org/doi/pdf/10.1145/3306618.3317950), as opposed to the larger list of terms from [(Smith et al., 2022)](https://arxiv.org/abs/2205.09209). We have included a clarification about that in our updated experiments section.
>
> **Do the columns in Table 1 (except for BA) represent a measure of fairness? The WR (word replacement) results seem quite performant. How do you justify the complexity of adding the style transfer and GPT3? What is the real advantage of the full $C$ dataset?**
>
> The rows in Table 1 represent CLP-trained classifiers using different sets of potentially similar pairs (generated by word replacement, style transfer, etc.). The columns represent different sets of similar pairs used to _evaluate_ individual fairness. For example, the last row/column indicates that training with CLP on the full $C$ dataset yields equal outcomes for 95.7% of pairs $(s,s')$ on a test set sampled from the same distribution as $C$.
>
> While the absolute gains in percentage points might look relatively small, fairness violations (i.e., pairs of actually similar sentences that are treated differently) represent concrete instances of model decisions being unduly influenced by demographic information and are examples of direct discrimination solely based on mentions of a particular demographic group.
>
> In Table 1 we observe that training on pairs generated by style transfer compared to our word replacement approach not just decreases the rate of such fairness violations on pairs generated by style transfer by more than a half from 9.5% to 4.6% but also decreases the rate on pairs generated by the other Word Replacement approach from [(Garg et al., 2019)](https://dl.acm.org/doi/pdf/10.1145/3306618.3317950) by more than a fourth from 6.9% to 4.7%. Further, the model trained on $C$ consistently yields the highest fairness results when cross-evaluated on a test distribution different from the one used for training.
>
> Part 1/2

---

> > ### Author Response · Authors · 2022-11-14
> > **Part 2**
> >
> > **The fairness metric that is used seems a bit ad hoc. One of the more standard metrics could be used, e.g., equalized odds, etc.**
> >
> > Given that different treatment of two similar comments is highly undesirable, as elaborated in the previous response, we believe that the frequency at which *human-validated* similar pairs are treated differently is an appropriate metric for measuring individual fairness, which is the main focus of our paper. We also would like to highlight that this metric is a natural upper bound for the certified fairness metric from ([Ruoss et al., 2020](https://proceedings.neurips.cc/paper/2020/file/55d491cf951b1b920900684d71419282-Paper.pdf); [Peychev et al., 2021](https://arxiv.org/abs/2111.13650)), as well as the prediction consistency metric from [(Yurochkin & Sun, 2020)](https://arxiv.org/abs/2006.14168), both of which consider equal predictions across all comments $s’$ similar to a given comment $s$ for simpler formal specifications of similarity. We clarified this point in the experimental section and now refer to our evaluation metric as “individual fairness” rather than “fairness” throughout the section.
> >
> > Further, we also added an additional experiment on equalized odds in App. F, finding that CLP training on the full set $C$ of generated pairs reduces the mean TNR gap between groups from 20.9 to 6.1 but increases the mean TPR gap from 5.8 to 15.7. When using the dataset  $\hat{C}^\star$ filtered based on predicted human fairness judgments for CLP training, the effect is less extreme with the mean TPR gap at 11.1 and the mean TNR gap at 10.0. These results are consistent with the results from Garg et al. (2019) who also find that improvements in individual fairness for toxicity classification coincide with better TNR gaps but worse TPR gaps.
> >
> > **"Fairness specifications" is a misleading term as it alludes to a theoretical formalism/specification. Similar pairs of text across protected groups is a more appropriate description of what the paper generates.**
> >
> > While literature on algorithmic fairness often focuses on encoding complex notions of fairness using simple mathematical equations that are convenient to work with, we believe that similar to other nuanced concepts like human intuition about the toxicity of comments, the correct notion of fairness is not expressible as a simple formula. Our results show that simple formal specifications of fairness in text classification, such as robustness to word replacement are insufficient to capture human fairness intuitions, both because they lack expressivity, and because they are unable to handle contextual nuances that often determine human fairness judgments. Correspondingly, in this work we specify (individual) fairness implicitly, via human-labeled data, similar to how labeled data is used to specify the ground truth for tasks like toxicity classification. We leave it to future work to explore extracting more human-readable specifications from classifiers trained on these labels.
> >
> > That said, we agree that the term “specifications” alludes to a more theoretical formalism, especially when paired with the word “intuitive” which we intended to point towards the better match with human intuitions achieved by collecting human fairness judgments. To address this issue we replaced the title with a more accurate one and clarified in the paper that our usage of the term “fairness specifications” refers to implicit specifications based on human judgments or approximations of these. If you think another title would be more suitable, please let us know.

---

### Author Response · Authors · 2022-11-14
**Response to Reviewers**

We thank the reviewers for their valuable and insightful feedback. We are glad they found the problem we study interesting ($\color{purple} 6aFV$), and important ($\color{red} Cvse$). We are encouraged that they found our approach novel ($\color{blue} cacL$,  $\color{red} Cvse$) and technically sound ($\color{blue} cacL$). We are also pleased that they see the potential for our method to be applied in additional contexts not studied in this paper ($\color{green} Y9Ec$).

We found the constructive feedback of the reviewers very helpful and have prepared an updated version of our manuscript. Additions to the manuscript are marked in blue in the revised manuscript and summarized below. We reply to each reviewer in more detail in individual responses.

* We have improved the clarity of our experiments section by providing additional details both in table captions as well as the main text to address concerns about clarity by reviewers $\color{green} Y9Ec$, $\color{red} Cvse$, and $\color{blue} cacL$). We have moved extended results on our human evaluation to Appendix A, in order to make space for this.
* We have replaced most references to our “fairness” metric with “individual fairness” and have added additional explanations of this metric to address further issues with clarity brought up by reviewers $\color{green} Y9Ec$, and $\color{red} Cvse$.
* We have added an experiment on group fairness (equality of odds) in App. F following requests by reviewers $\color{green} Y9Ec$ and $\color{blue} cacL$
* We have update our paper’s title to “Human-Guided Fair Classification for Natural Language Processing” to address reviewer $\color{green} Y9Ec$’s concerns about the term “fairness specifications” in our initial title, and have added clarification on what we mean by that term in the main text.
* Based on reviewer $\color{purple} 6aFV$’s comment, we have further highlighted throughout the paper that our approach requires a task-specific instantiation and that our experiments focus on the instantiation to the task of toxicity classification.
* We have also extended our ethical considerations section with additional discussion on disagreement between human annotators and geographical bias in our survey participants to address further concerns of reviewer $\color{purple} 6aFV$.
* Following the suggestion by reviewer $\color{blue} cacL$, we have replaced references to the better “diversity” of our generated pairs compared to word replacement with references to better “coverage”.
* Based on reviewer $\color{red} Cvse$’s request, we have added a table of example generations by our different method using the same source sentence and target demographic attribute (Table E.5)

---

### Author Response · Authors · 2022-11-18
**Further Discussion**

Dear Reviewers,

The main discussion period is coming to an end and we hope that we have addressed all of your concerns. Please, let us know if you have any follow-up questions. We will be happy to answer them.

Kind regards,\
the Authors

---

### Decision · Program_Chairs · 2023-01-20

**Decision:**

Accept: notable-top-25%

**Justification For Why Not Higher Score:**

The spotlight is recommended based on the reviewers' comments and ratings towards "novelty and significance". In addition, there are a few weaknesses pointed out by reviewers such as task dependence and lack of clarity towards certain design choices, and broad impact/limitation discussion. All of these led to a "spotlight" recommendation, rather than an "oral".



**Justification For Why Not Lower Score:**

Reviewers all agreed that this work focuses on a very important research question, and is technically solid with comprehensive experiments. All of these make it suitable for a spotlight representation, rather than a poster.

**Metareview: Summary, Strengths And Weaknesses:**

This work introduces a novel methodology for generating data pairs of semantically similar sentences that differ along sensitive attributes, with a case study of toxicity classification. Reviewers all agreed that this work is technically solid and novel with comprehensive experiments; it focuses on a very important research question, and the writing is clear and easy to follow. Reviewers all recommend acceptance. I’d like to encourage the reviewers to incorporate reviewers’ suggestions to improve the revised version in terms of some clarity and broad impact/limitation discussion.


**Note From Pc:**

if the above contains the word "oral" or "spotlight" please see: "oral" presentation means -> notable-top-5% and "spotlight" means -> notable-top-25%. As stated in our emails, we are disassociating presentation type from AC recommendations

**Summary Of Ac-Reviewer Meeting:**

N/A